# Spin-orbit-splitting-driven nonlinear Hall effect in NbIrTe$_4$

Ji-Eun Lee [1,2,3,4], Aifeng Wang [5,6], Shuzhang Chen [5,7], Minseong Kwon [2,8], Jinwoong Hwang [1,9], Minhyun Cho[8], Ki-Hoon Son[2], Dong-Soo Han [2], Jun Woo Choi [2], Young Duck Kim [8], Sung-Kwan Mo [1], Cedomir Petrovic [5,7,10], Choongyu Hwang [3] ✉, Se Young Park [11,12] ✉, Chaun Jang [2] ✉ & Hyejin Ryu [2] ✉

The Berry curvature dipole (BCD) serves as a one of the fundamental contributors to emergence of the nonlinear Hall effect (NLHE). Despite intense interest due to its potential for new technologies reaching beyond the quantum efficiency limit, the interplay between BCD and NLHE has been barely understood yet in the absence of a systematic study on the electronic band structure. Here, we report NLHE realized in NbIrTe$_4$ that persists above room temperature coupled with a sign change in the Hall conductivity at 150 K. First-principles calculations combined with angle-resolved photoemission spectroscopy (ARPES) measurements show that BCD tuned by the partial occupancy of spin-orbit split bands via temperature is responsible for the temperature-dependent NLHE. Our findings highlight the correlation between BCD and the electronic band structure, providing a viable route to create and engineer the non-trivial Hall effect by tuning the geometric properties of quasiparticles in transition-metal chalcogen compounds.

Berry curvature (BC) is a key to understand novel physical phenomena such as anomalous Hall effect, chiral anomaly, topological Hall effect, and spin-valley Hall effect. Moreover, the BC classifies the topology of a solid via a topological number that predicts the presence of protected states at its boundary. The response of a system governed by the BC is constrained by the Onsager relation in the linear order[1]. This stringent constraint, however, is no longer valid in the high-order responses that are proportional to the second-order or even higher-orders of the driving field. The integration over higher-order fields gives a nonlinear response of the system, contributing to the optical and transport responses[2–5]. This finding not only provides a methodology to explore the momentum texture of the BC of a system, but also paves a way to utilize the response even though the linear order is vanishingly weak or when a large driving field makes the higher-order response exceed the linear order. To make use of the higher-order response has great potential for applications in rectification devices[6,7], photosensitive devices[8], and photovoltaic devices[9] that potentially overcome the quantum efficiency limit[10,11].

[1]Advanced Light Source, Lawrence Berkeley National Laboratory, Berkeley, CA 94720, USA. [2]Center for Spintronics, Korea Institute of Science and Technology (KIST), Seoul 02792, South Korea. [3]Department of Physics, Pusan National University, Busan 46241, South Korea. [4]Max Planck POSTECH Center for Complex Phase Materials, Pohang University of Science and Technology, Pohang 37673, South Korea. [5]Condensed Matter Physics and Materials Science Department, Brookhaven National Laboratory, Upton, New York 11973, US. [6]Low Temperature Physics Laboratory, College of Physics and Center of Quantum Materials and Devices, Chongqing University, Chongqing 400044, China. [7]Department of Physics and Astronomy, Stony Brook University, Stony Brook, New York 11794-3800, USA. [8]Department of Physics and Department of Information Display, Kyung Hee University, Seoul 02447, South Korea. [9]Department of Physics and Institute of Quantum Convergence Technology, Kangwon National University, Chuncheon 24341, South Korea. [10]Shanghai Advanced Research in Physical Sciences, Shanghai 201203, China. [11]Department of Physics and Origin of Matter and Evolution of Galaxies (OMEG) Institute, Soongsil University, Seoul 06978, South Korea. [12]Integrative Institute of Basic Sciences, Soongsil University, Seoul 06978, South Korea. ✉e-mail: ckhwang@pusan.ac.kr; sp2829@ssu.ac.kr; cujang@kist.re.kr; hryu@kist.re.kr

Recent studies on the higher-order response have invited nonmagnetic materials with broken centrosymmetry as a new member of the Hall effect family, so-called nonlinear Hall effect (NLHE), characterized by a quadratic behavior of the Hall voltage with second-harmonic ($2\omega$) frequencies in the presence of a perpendicular AC driving current. In the absence of the linear Hall effect due to time-reversal symmetry, the lowest-order Hall current is driven by the Berry curvature dipole (BCD)[4]. Since the response is proportional to the gradient of the BC, tilted anticrossing bands and Weyl points[12–14] are predicted to exhibit strong BCD that can generate a nonlinear Hall angle close to 90 degrees[15]. As a result, the momentum-dependent texture of the BC based on the electronic structures is essential to understand the NLHE in which the evolution of BC momentum texture by lattice strain, interlayer twisting, and external electric fields[15] can lead to NLHE-based device applications[6,16–22]. Nevertheless, while the majority of studies on the NLHE have predominantly focused on exploring its transport properties, applications, and theoretical simulations, the direct experimental confirmation and comprehensive understanding of the BCD based on electronic structures have not yet been fully attained. This achievement would provide crucial insights into the underlying mechanism and controllability of the BCD.

In this paper, we report the room temperature NLHE in NbIrTe₄ thin flakes, which exhibit a frequency-doubled Hall conductivity proportional to the square of the driving current. We also demonstrate that the sign change in the NLHE at 150 K is induced by the sign change

of the BCD because of the chemical potential shift at high temperatures. It is unambiguously evidenced by direct observation of a chemical potential shift in the temperature-dependent band dispersion using angle-resolved photoemission spectroscopy (ARPES) and calculated BCDs. Investigation of the electronic structures using ARPES and density functional theory (DFT) also indicates that the main contributor of BCDs is a partial occupation of spin–orbit split bands. Our findings provide important insights into the momentum texture of the Berry curvature and into controlling the Berry curvature dipole hosting the NLHE, which can be utilized for NLHE-based devices.

## Results and discussion

### Prediction of the nonlinear Hall effect in NbIrTe₄

The crystal structure of bulk NbIrTe₄ (Fig. 1a) has an orthorhombic unit cell with space group $Pmn2_1$[23]. The experimental lattice constants are $a = 3.77\,\text{Å}$, $b = 12.51\,\text{Å}$, and $c = 13.12\,\text{Å}$, where the two-dimensional planes are stacked along the $c$-axis. The material has a nonmagnetic metallic phase, and no anomalous Hall effect is expected. Moreover, because of the combination of mirror ($\{M_a \mid (0,0,0)\}$) and glide mirror ($\{M_b \mid (1/2,0,1/2)\}$) operations, no NLHE is expected within the $ab$ plane[23–25]. However, in the slab geometry, the symmetry is lowered to the $Pm$ space group due to the breaking of the translation along the $c$-axis, leaving only identity and mirror ($\{M_a \mid (0,0,0)\}$) operations (Fig. 1b). The resultant symmetry allows nonlinear Hall current along the $b$-axis when there is a driving current along the $a$-axis, parallel to

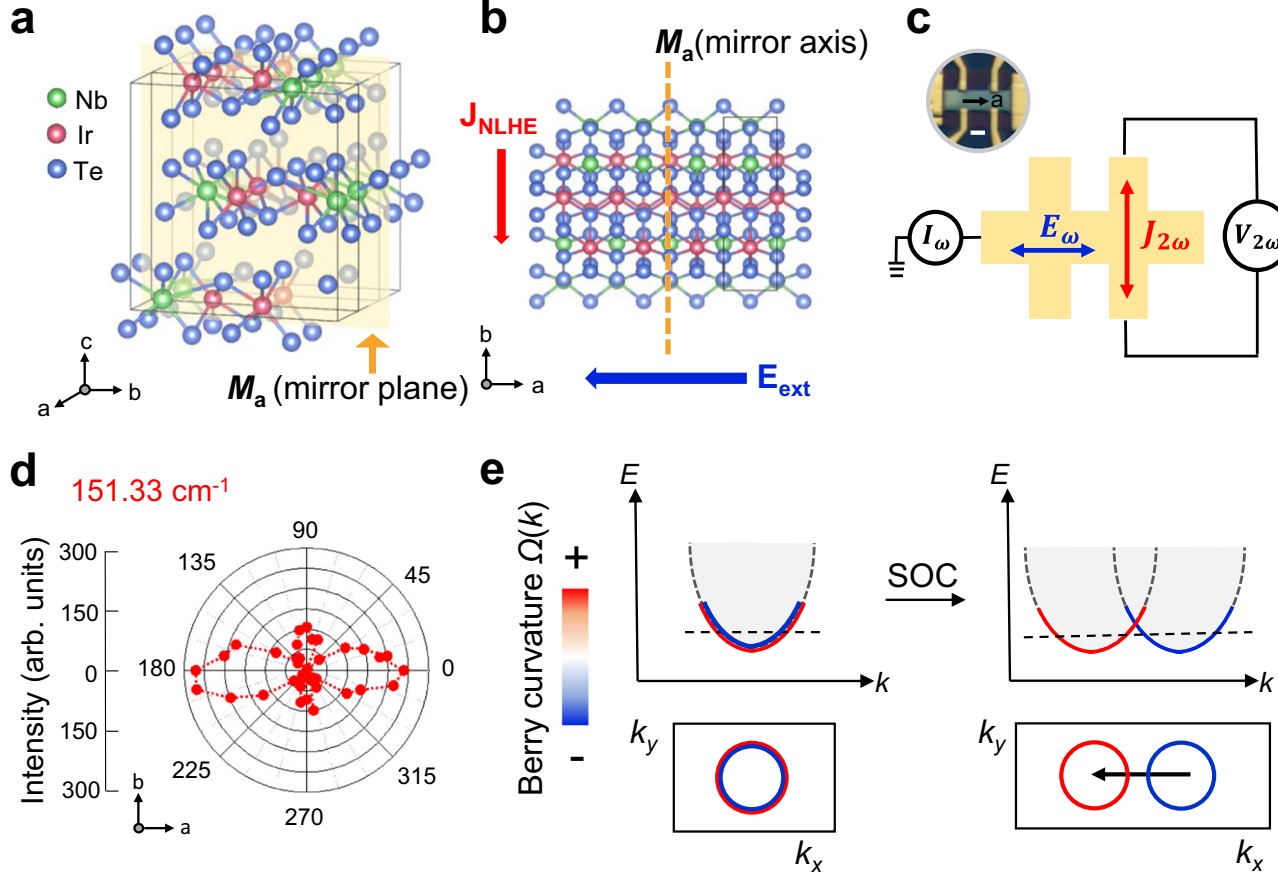

**Fig. 1 | Crystal structure symmetry and band model of NbIrTe₄.** **a** Crystal structure of NbIrTe₄ exhibiting broken inversion symmetry with a mirror plane ($M_a$) illustrated as a yellow plane. Top view (**b**) of the crystal structure of NbIrTe₄ with mirror axis $M_a$ (yellow dashed line). $J_{NLHE}$ is the generated nonlinear Hall current parallel to $M_a$, and $E_{ext}$ is the applied electric field perpendicular to $M_a$.
**c** Illustrations of Hall devices with an optical image (top left inset). $I_\omega$, $V_{2\omega}$, $E_\omega$, and $J_{2\omega}$ represent the current, voltage, electric field, and generated nonlinear Hall current,

respectively, at frequencies $\omega$ and $2\omega$. The scale bar in the optical image (top left inset) is 1 μm. **d** Polar plots of Raman modes at 151.33 cm⁻¹. The results are shown as intensity versus angle configuration. **e** A schematic model picture of band structures with Berry curvature (BC) $\Omega(k)$ evolution, including spin–orbit coupling (SOC). The lower panels display momentum-resolved BC, highlighting the emergence of a large dipole hotspot of BC in the presence of SOC.

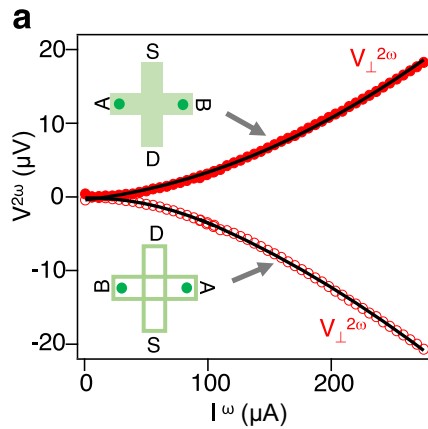
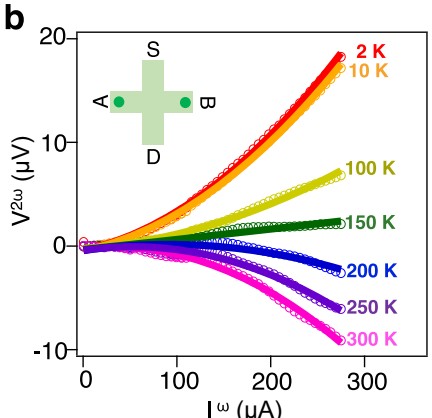

**Fig. 2 | Nonlinear Hall response in NbIrTe₄. a** Second-harmonic $V_\perp^{2\omega}$ as a function of the AC current amplitude $I^\omega$ scaling quadratically in a 15-nm-thick flake of NbIrTe₄ at 2 K. The black solid line is a quadratic fit to the data. The current is along the a-axis. Green cross bars in the inset represent the geometry of the measurements. The applied current is injected from the source (S) electrode to the drain (D)

electrode, and the voltage is measured between the A and B electrodes. A sign change occurs upon simultaneous reversal of both the applied current direction and the corresponding Hall probes. **b** Temperature dependence of the NLHE and a sizable NLH signal at room temperature, with a sign change at ~150 K.

the direction of the BCD (Fig. 1b)[24,26]. The devices were fabricated into a Hall bars pattern (Fig. 1c), which is aligned in the crystallographic direction by angle-resolved polarized Raman spectroscopy (Fig. 1d). Crystallographic directions were confirmed via high intensity along the a-axis and low intensity along the b-axis, which is evidence of broken inversion symmetry in NbIrTe₄, consistent with the previous report[27]. In this system, the interplay between spin–orbit coupling and the presence of broken inversion symmetry leads to the emergence of a prominent BCD hotspot within spin–orbit-split bands (Fig. 1e), as elaborated upon in the subsequent theoretical section.

## Observation of nonlinear Hall effect in NbIrTe₄

To investigate the NLHE in NbIrTe₄, nonlinear transport measurements on a 15-nm-thick Hall bar device fabricated from a NbIrTe₄ flake were performed under zero magnetic field (Fig. 2). The second-harmonic transverse voltage ($V_\perp^{2\omega}$) under zero magnetic field in a 15-nm-thick NbIrTe₄ flake at 2 K responds quadratically to the current $I^\omega$ along the a-axis, indicating the presence of the NLHE on NbIrTe₄ flakes (Fig. 2a). Additionally, we have verified both the direction and frequency of driving AC current-dependent nonlinear Hall responses (see Supplementary Figs. 2, 3), thereby confirming the absence of experimental measurement artifacts. The second-harmonic transverse voltage ($V_\perp^{2\omega}$) of a NbIrTe₄ flake gradually decreases as the temperature increases from 2 to 150 K (nearly decay), followed by a sign change with an increasing magnitude as the temperature further increases to 300 K (Fig. 2b), consistent with the NLHE observed in TaIrTe₄[6]. However, the slight difference in the sign-changing temperature and the magnitude of NLHE between the two materials are due to the band structure changes associated with increased ionic radius and the spin–orbit coupling strength from Nb to Ta[12,15].

## Temperature-dependent chemical potential shift nature of NbIrTe₄

Since the NLHE is derived from BCD, understanding the nature of BCD from the perspective of electronic band structures enables both understanding and effective control of the NLHE. To investigate the electronic bands that dominantly contribute to NLH behavior, we explored the electronic band structure using ARPES experiments and DFT calculations. The Fermi surface of a NbIrTe₄ single crystal consists of elliptical-shaped electron pockets around the $\bar{\Gamma}$ point (A in Fig. 3b) and semi-elliptical-shaped broad bulk hole pockets along the $\bar{\Gamma}$-$\bar{X}$ directions (B in Fig. 3b). Both Fermi surface

features (Fig. 3b) and the band dispersions along the $\bar{X}$-$\bar{\Gamma}$-$\bar{X}$ and $\bar{S}$-$\bar{Y}$-$\bar{S}$ directions (Fig. 3c) are in good agreement with the DFT calculation results shown by red dotted lines in Fig. 3b, c and those reported previously[8,25,28–30].

To further elucidate the origin of the temperature dependence of the NLHE behavior (Fig. 2b), ARPES spectra along the $\bar{X}$-$\bar{\Gamma}$-$\bar{X}$ direction were acquired at various temperatures (Supplementary Fig. 6). The energy distribution curves (EDCs) at the $\bar{\Gamma}$ point as a function of temperature (Fig. 3d) demonstrate that, as the temperature increases, the two peaks at $E - E_F = -0.3$, $-1.0$ eV of the EDCs shifts systematically to lower binding energy. In Fig. 3e, we show the temperature-dependent energy shift ($\Delta E$) for the peaks of EDCs at the $\bar{\Gamma}$ point (blue, $\Delta E = E - E_{T=20\,K}$), obtained from multi-peak fits with multiplication of Fermi–Dirac distribution (FD) function and convolution of instrumental resolution as detailed in Fig. S7. It is clearly observed that the $\Delta E$ shifts about 15–20 meV as the temperature increases from 10 to 280 K, which implies the chemical potential shifts down to higher binding energy, which is consistent with our simulated ARPES spectra (See Supplementary Fig. 11). The increase of the hole pockets in the Fermi surface associated with the chemical potential shift is consistent with the behavior of the hole carrier density, increasing as a function of temperature obtained from the reported transport results[29,31]. The ARPES and transport results suggest that there is a substantial change in the band structures with increasing temperature. This could be related to the sign change in BCD, which suggests further investigation through DFT analysis.

## Origin of the BCD of NbIrTe₄

We investigated the mechanism that induces the nonlinear Hall conductivity and the origin of the sign change by first-principles calculations. The current induced by the NLHE in two-dimensional systems is expressed as[4]

$$\mathbf{j}(2\omega) = \frac{e^3 \tau}{2\hbar^2(1 + i\omega\tau)}\hat{\mathbf{z}} \times \mathbf{E}(\omega)(\mathbf{D}(\omega) \cdot \mathbf{E}(\omega)) \tag{1}$$

where $\hbar$ is Planck's constant, $\hat{\mathbf{z}}$ is the direction normal to the plane parallel to the c-axis, $\tau$ is the scattering time, $\mathbf{E}(\omega)$ is the external electric field, and $\mathbf{D}(\omega)$ is the BCD vector. Given the symmetry of the NbIrTe₄ slab (space group $Pm$), $\mathbf{D}(\omega)$ is written as $D_a(\omega)\hat{\mathbf{a}}$, where $\hat{\mathbf{a}}$ is the unit vector along the a-axis, which gives the nonlinear current in the b-axis under electric fields applied along the a-axis. $D_a(\omega)$ is obtained by the

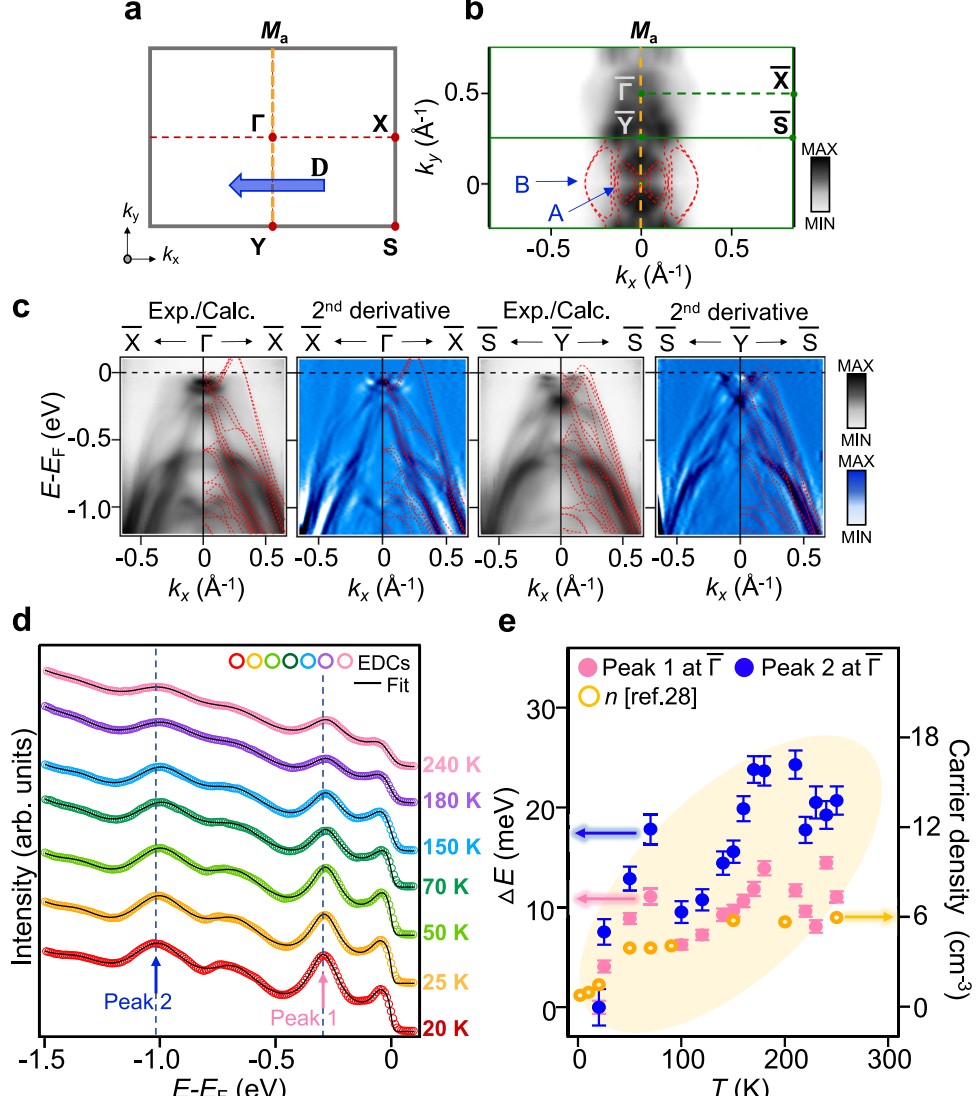

**Fig. 3 | Electronic structures and evidence for the temperature-dependent chemical potential shift of NbIrTe₄.** **a** The surface Brillouin zone of NbIrTe₄ with high-symmetry points marked as red points. The blue arrow represents the BCD (D). **b** ARPES and calculated (red dotted lines) Fermi surface (FS) of NbIrTe₄ through two Brillouin zones (green solid lines) at 20 K. $M_a$ is a mirror plane. **c** ARPES intensity plots with the calculated band structure (red dotted lines) and corresponding second-derivatives ARPES spectra for the enhanced visibility along the $\overline{X}$-$\overline{\Gamma}$-$\overline{X}$ and $\overline{S}$-$\overline{Y}$-$\overline{S}$ directions at 20 K. **d** The energy distribution curves (EDCs) at the $\overline{\Gamma}$ point as a function of temperature along with black fitted curves. The multi-peak

fits are obtained by multiplication of the Fermi−Dirac distribution function and convolution of instrumental resolution with Lorentzian curves (detailed in Fig. S7). Blue dotted lines are the position of the two EDC peaks at $E - E_F = -0.3, -1.0$ eV, respectively (where $E_F$ is the Fermi energy). **e** Temperature dependence of the energy shifts $\Delta E$ (left axis) are taken from the peak position shift of EDCs at $\overline{\Gamma}$ ($\Delta E = E - E_{T=20\,K}$) point obtained by multi-peak fit (see Supplementary Fig. 7). Hole carrier density ($n$, right axis) is obtained from the reported Hall measurement[31]. All error bars are defined as the standard deviation of fitting the position of the peaks.

derivative of the out-of-plane BC along the *a*-axis, expressed as

$$D_a(\omega) = \sum_n \int_{BZ} \frac{d^2\mathbf{k}}{4\pi^2} f(\hbar\omega - \epsilon_{n\mathbf{k}}) \frac{\partial \Omega_{n\mathbf{k},c}}{\partial k_a}, \qquad (2)$$

where *n* is the band index, $\epsilon_{n\mathbf{k}}$ denotes the single-particle energy, $f(\hbar\omega - \epsilon_{n\mathbf{k}})$ is the Fermi−Dirac distribution function, and $\Omega_{n\mathbf{k},c}$ is the Berry curvature along the *c*-axis. From the expression of BCD, the important parts of the Brillouin zone are regions with a large change in the BC along the *a*-axis. To identify the characteristics of the band structures that are the main contributors to the BCD, we first calculate $D_a(\omega)$ for a bilayer NbIrTe₄ slab. Compared with a thick slab geometry, which is difficult to analyze because of dense subbands, the bilayer system having a similar energy dependence of

BCD with 12-layer NbIrTe₄ is suitable to pinpoint the hotspots of BC that mainly contribute to the BCD.

We find that the BCD is dominantly contributed from the partially occupied bands with spin−orbit induced splitting by analyzing the momentum texture of BC, integrated around the energy at which BCD changes abruptly. Figure 4a shows the energy dependence of $D_a(\omega)$, where a large increase in $D_a$ is observed at around −10 meV. The momentum-resolved BC integrated in the energy range from −12 to −7 meV (red shaded area in Fig. 4a) shows that the BC is concentrated in small areas around the Γ point, with a clear sign change in $\Omega_{n\mathbf{k},c}$ across the $M_a$ mirror axis (Fig. 4b), which dominantly contributes to a sharp decrease in $D_a$. The bands contributing to the BC around −10 meV are presented in Fig. 4c, which corresponds to the region indicated by the black-dotted square in Fig. 4b. The one-dimensional

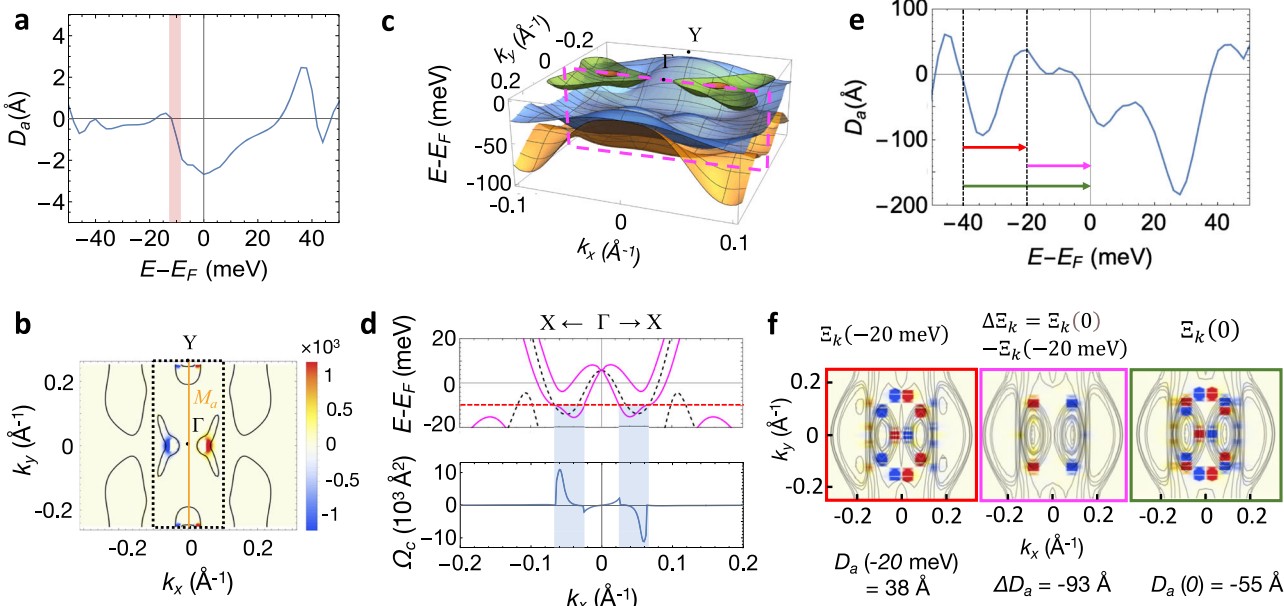

**Fig. 4 | Calculated Berry curvature dipole and momentum-dependent Berry curvature. a** The BCD ($D_a$) contributing to the in-plane NLHE for bilayer NbIrTe$_4$, plotted as a function of the chemical potential. **b** Momentum-resolved BC ($\Omega_c(\mathbf{k})$ in Å$^2$) of bilayer NbIrTe$_4$ integrated from −12 to −7 meV corresponding to the red shaded area in panel (**a**). **c** Band structures in the area denoted as the black-dotted square in panel (**b**). **d** Band dispersion around the $k$-points contributing large Berry curvatures of bilayer NbIrTe$_4$. The top panel shows bands near the Fermi energy expressed as solid magenta lines along the $\bar{X}$-$\bar{\Gamma}$-$\bar{X}$ cut (dotted magenta box in panel (**c**). The black dashed lines are the bands calculated without spin−orbit coupling. The bottom panel is $\Omega_c(\mathbf{k})$ integrated up to −10 meV relative to $E_F$ (red dashed line). **e** Chemical potential versus the component of the $D_a$ contributing to the in-plane NLHE for 12-layer NbIrTe$_4$. **f** Integrated momentum-resolved BC ($\Xi_\mathbf{k}(E)$) for 12-layer NbIrTe$_4$ (arb. units) integrated from −40 meV to $E_F$, with the left, middle, and right panels corresponding to the energy range marked with red, magenta, and green arrows in panel (**e**), respectively. The black lines denote iso-energy surfaces at $E_F$ − 20 meV (left), $E_F$ − 10 meV (middle), and $E_F$ (right).

cut along the $\bar{X}$-$\bar{\Gamma}$-$\bar{X}$ line reveals the characteristics of these bands (Fig. 4d), where large Rashba spin−orbit splitting is observed, consistent with the absence of inversion symmetry. The BC calculated along the $\bar{X}$-$\bar{\Gamma}$-$\bar{X}$ line (bottom panel of Fig. 4d) shows that partially occupied spin-split bands from spin−orbit coupling contribute largely to the BC ($\Omega_c$) (blue shaded area in Fig. 4d). Thus, we attribute the large change in the BCD to a shift of the $E_F$ as the occupation of the spin−orbit split bands changes accordingly. We note that the BCD and momentum-resolved BC of the bilayer slab are insensitive to the choice of the local density approximation (LDA) or generalized gradient approximation (GGA) for the exchange-correlation potential (see Supplementary Fig. 12).

The sign change around the Fermi energy of the calculated BCD in a thicker slab (Fig. 4e) provides clear evidence that the sign inversion of the Hall conductivity can be induced by the Fermi energy shift. The calculated BCD for a 12-layer slab, which is sufficiently thick to represent the experimental geometry (see Supplementary Fig. 8), shows a nonzero value of about $D_a = −60$ Å at the Fermi energy. More importantly, the BCD changes sign as $E − E_F$ decreases, showing a positive peak about $D_a = 40$ Å around −20 meV, which is comparable to the peak shift observed in the ARPES results (Fig. 3d). Thus, we propose that the observed sign change in the NLHE with increasing temperature is induced by a negative chemical potential shift, as evidenced by a sign change in the calculated BCD. Reducing the number of electrons corresponding to a 0.025 h/f.u. doping induces the chemical potential shifts about −20 meV both for bulk and slab geometry (see Supplementary Fig. 12). We note that the $D_a$ from the experimental data is estimated to be −348 Å at low temperatures (see Supplementary Note 8), which is similar to the theoretical value of −55 Å, as explained in the following discussion.

Figure 4f shows the change in the momentum texture of the BC of the 12-layer slab responsible for the energy-dependent sign change in BCD, where the dominant contribution to the change in the BC

originates from small concentrated areas in the Brillouin zone, similar to the bilayer case. Since the BCD is nearly zero at −40 meV, we define the integrated BC in the energy interval from −40 meV to $E_F$ as $\Xi_\mathbf{k}(E) = \sum_n \int_{E_F-40\,\mathrm{meV}}^{E_F+E} dE' \delta(E' - \epsilon_{n\mathbf{k}}) \Omega_{n\mathbf{k},c}$. Since the BCD along the $a$-axis $D_a(E)$ is proportional to $\frac{\partial \Omega_{n\mathbf{k},c}}{\partial k_a}$ summed over the bands below the energy $E$, large BCD is expected around the $k$-points showing an abrupt change in $\Xi_\mathbf{k}(E)$ when integrated over a narrow energy range. Thus, $\Xi_\mathbf{k}(−20\,\mathrm{meV})$, $\Delta\Xi_\mathbf{k} = \Xi_\mathbf{k}(0) - \Xi_\mathbf{k}(−20\,\mathrm{meV})$, and $\Xi_\mathbf{k}(0)$ (corresponding to red, magenta, and green arrows Fig. 4e, respectively) represent the momentum-dependent BC contributing $D_a$ at −20 meV (red box in Fig. 4f), $\Delta D_a$ between −20 meV and $E_F$ (magenta box), and $D_a$ at $E_F$ (green box), respectively. The positive BCD at −20 meV can be explained on the basis of an overall negative-to-positive sign change in the integrated BC across the mirror axis, giving a net positive value. The sign of BCD changes around $E_F$ as a result of the addition of the opposite component of the BCD shown in the middle panel with the opposite sign change in the BC across the mirror axis. Thus, the BCD at $E_F$ has a net negative sign with the BC texture, which is the sum of those corresponding to $D_a$ at −20 meV and $\Delta D_a$. From the analysis of the bilayer case, we expect that the BCD of a 12-layer slab can also be mainly contributed by the partially occupied spin-split bands. Our analysis reveals the mechanism of the temperature-dependent sign change in the NLHE observed in the transport measurements. The shift in $E_F$, as observed in the temperature-dependent ARPES measurements, changes the distribution of the BC sensitive to band filling, resulting in the opposite sign of the BCD as the temperature increases.

We report the NLHE in nonmagnetic NbIrTe$_4$ flakes. Our transport measurements show that the magnitude of the Hall voltage at a doubled frequency increases proportional to the square of the driving current, indicating the occurrence of the NLHE. Moreover, the measured Hall voltage shows a sign change with increasing temperature

that persists up to room temperature. ARPES measurements and first-principles calculations reveal that the BCD in NbIrTe$_4$ originates mainly from partially filled spin−orbit split bands. Furthermore, we identified the mechanism of the sign changes in the NLHE on the basis of the chemical potential shift and associated change in the distribution of the BC in momentum space, which is supported by temperature-dependent energy shift in the ARPES results and sign-changing BC texture in the slab calculation. The identified mechanism suggests that the NLHE can be electronically controlled by changing the momentum-dependent texture of the BC, which can be used to enhance the efficiency of BCD-based devices such as memories and rectifiers.

## Methods

### Device fabrication and electrical measurements

Single crystals of NbIrTe$_4$ were grown using the flux technique as described elsewhere[32] and as follows. Stoichiometric amounts of Nb (99.999%) and Ir (99.99%) along with excess Te (ratio of Nb: Ir: Te = 1: 1: 20) were heated within a sealed quartz ampule, reaching temperatures up to 1000 °C before gradually cooling to 700 °C. Subsequently, the excess Te was eliminated by centrifuging the ampules at 700 °C. NbIrTe$_4$ samples were mechanically exfoliated from a single crystal NbIrTe$_4$ and transferred onto SiO$_2$/Si substrates. Hall bar device geometries were patterned with Ti(10 nm)/Au(100 nm) electrodes on NbIrTe$_4$ crystals. To remove the oxidation layer on the surface of the NbIrTe$_4$ flakes, Ar$^+$-ion sputtering at 200 V was employed for 300 s. To perform nonlinear transport measurements, the device was biased with a harmonic current along the $a$-axis at a fixed frequency ($\omega$ = 117.77 Hz) and measured both the first-harmonic frequency ($\omega$) and second-harmonic frequency ($2\omega$) using a lock-in amplifier.

### Raman spectroscopy

A linearly polarized 514 nm laser was focused to a spot of approximately 1–2 μm onto the nanoflakes at room temperature. The laser power was limited to less than 200 μW. A polar plot of angle-resolved measurements was obtained by fixing the Raman modes at 151.3 cm$^{-1}$ and comparing the intensity as a function of angle[33].

### ARPES measurements

ARPES measurements were performed at Beamline 10.0.1, Advanced Light Source, Lawrence Berkeley National Laboratory. The ARPES system was equipped with a Scienta R4000 electron analyzer. The photon energy was set to be 60 eV, with an energy resolution of 20 meV and an angular resolution of 0.1 degrees.

### First-principles calculations

We used the first-principles DFT to calculate the electronic structures and nonlinear Hall conductivity. The calculations were performed using the Vienna Ab-initio Simulation Package (VASP). The Ceperley−Alder (CA)[34,35] and Perdew−Burke−Ernzerhof (PBE)[36,37] parameterizations were used for the local density approximation (LDA) and the generalized gradient approximation (GGA), respectively. The projector augmented-wave method[38] was used with an energy cut-off of 500 eV. Spin−orbit coupling was included. For bulk and slab NbIrTe$_4$, $k$-point sampling grids of $3 \times 12 \times 3$ and $3 \times 12 \times 1$ were used, respectively. The experimental atomic structures were used to construct the bulk and slab geometry of 2- and 12-layer NbIrTe$_4$ with a vacuum layer of 16 Å. For the calculations of the BC and the BCD, the Wannier90 code[39] was used to construct the tight-binding Hamiltonian using Nb-$d$, Ir-$d$, and Te-$p$-derived bands. For the bilayer, the tight-binding Hamiltonian was constructed from the bilayer-vacuum configuration. For the thicker slab, the tight-binding Hamiltonian was constructed by making a supercell of the bulk tight-binding Hamiltonian using the Python Tight Binding (PythTB) code[40]. The slab band structures of the supercell tight-binding Hamiltonian and those of DFT using vacuum-slab geometry were found to be in good agreement

(Supplementary Fig. 11). The calculations with a small hole doping of 0.025 h/f.u. were done by reducing the number of electrons with a compensating uniform background charge. The BC and BCD were calculated using the Wannier−Berri code[41,42] and the surface bands were calculated using WannierTools code[43].

## Data availability

All data generated in this study are provided in the article and Supplementary Information. Additional data and materials are available from the corresponding authors upon request.

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

## Acknowledgements

J.-E.L., K.-H.S., D.-S.H., J.W.C., C.J. and H.R. were supported by the KIST Institutional Program (2E32251, 2E31542), and the National Research Foundation of Korea (NRF) grant funded by the Korea government (MSIT) (No. 2021R1A2C2014179, 2020R1A5A1016518, 2021R1A2C2011007, 2021M3H4A1A03054856, 2022M3H4A1A04096396, 2022M3H4A1A04074153, and RS-2023-00284081). A.W., S.C., and C.P. were supported by the US DOE-BES, Division of Materials Science and Engineering, under Contract No. DESC0012704 (BNL). The ARPES experiments performed at the ALS by S.-K.M. and J.-E.L. were supported by the Office of Basic Energy Sciences, US DOE, under contract No. DE-AC02-05CH11231. J.-E.L. was supported by in part by an ALS collaborative Postdoctoral Fellowship and the Max Planck POSTECH/Korea Research Initiative funded by the National Research Foundation of Korea (NRF) (2022M3H4A1A04074153). C.H. acknowledges support from the NRF grant funded by MSIT (RS-2023-00221154 and 2021R1A2C1004266) and the National Research Facilities and Equipment Center (NFEC) grant funded by the Ministry of Education (2021R1A6C101A429). C.J. acknowledges support from the Institute of Information & communications Technology Planning & Evaluation (IITP) grant funded by MSIT (2022-0-01026). S.Y.P. acknowledges support from the NRF grant funded by MSIT (2021R1C1C1009494 and RS-2024-00358551) and from the Basic Science Research Program through the NRF funded by the Ministry of Education (2021R1A6A1A03043957 and 2021R1A6A1A10044154). M.K., M.C., and Y.D.K. acknowledge support from NRF of Korea (2021K1A3A1A32084700 and 2022R1A4A3030766). Raman spectroscopy measurements were supported by the Korea Basic Science Institute (National Research Facilities and Equipment Center) grant funded by the Ministry of Education (2021R1A6C101A437). J.H. was supported by the National Research Foundation of Korea (NRF) grant funded by the Korean government (MSIT) (RS-2023-00280346). The Global Research Development Center (GRDC) and the Cooperative Hub Program through the NRF are funded by the Ministry of Science and ICT (MIST) (RS-2023-00258359). C.P. acknowledges support from the Shanghai Key Laboratory of Material Frontiers Research in Extreme Environments, China (No. 22dz2260800) and Shanghai Science and Technology Committee, China (No. 22JC1410300).

## Author contributions

J.-E.L., C.H., S.Y.P., C.J. and H.R. proposed and designed the research. Y.L. and C.P. performed single-crystal growth. J.-E.L., M.K. and M.C. performed the transport and Raman spectroscopy measurements and analyzed the data with assistance from D.-S.H., Y.D.K., C.J. and H.R.; J.-E.L., J.H. and H.R. carried out the ARPES measurements and analyzed the ARPES data with assistance from C.H. and S.K.M.; S.Y.P. carried out the density functional calculations and provided theoretical support. J.-E.L., S.Y.P., C.J. and H.R. wrote the manuscript and revised it with assistance from K.-H.S., C.H., J.W.C., C.P. and S.K.M. All authors contributed to the scientific planning and discussions.

## Competing interests

The authors declare no competing interests.
