## [Peer Review File · Nature Communications]

REVIEWER COMMENTS

Reviewer #1 (Remarks to the Author):

The authors reported the observation of the nonlinear Hall effect (NLHE) in NbIrTe₄ thin flakes, attributed to the Berry curvature dipole (BCD) from the broken inversion symmetry while the time-reversal symmetry is preserved. Combining ARPES and DFT calculations, the authors concluded that the BCD is governed by the momentum texture of Berry curvature hotspots from partially occupied spin-orbit split bands, providing an explanation for the temperature dependence of the NLHE signal—especially the occurrence of a sign change.

The results presented in the manuscript represent a relatively thorough investigation of the NLHE in NbIrTe₄. However, I am not convinced how this work represents a significant step forward compared to the established literature on similar topics—see, e.g., Ref. 6 in the manuscript, which addresses the NLHE in TaIrTe₄. The authors simply stated that “Most of the studies on NLHE, however, have focused on transport properties and their applications.” From my understanding, the methodology adopted and the main physical implications in Ref. 6 are similar to that reported here in this work—except for a lack of ARPES evidence for the chemical potential shift with temperature in Ref. 6. The energy- and momentum-resolved distribution of the BCD from calculations and the correlation with the NLHE have been discussed for TaIrTe₄. Therefore, the results presented in this manuscript for NbIrTe₄ certainly add to the existing knowledge about the NLHE but seem to be not of sufficient significance and novelty to be published in Nature Communications.

Minor comments:

1. Since the manuscript focuses on the NLHE, the completeness of the manuscript would benefit from the presentation of more detailed characterization of the NLHE: e.g., by showing the absence of the NLHE when the current is applied along directions other than the *a* axis, showing the absence of dependence on the frequency of the current (while maintaining the double-frequency condition), by performing scaling analysis, etc. These details would further corroborate the NLHE nature of the observed transport behavior, help to exclude measurement artifacts, and shed light on its origins. This is merely a suggestion.

2. Line 231-232 of the manuscript: “which is similar to the theoretical value of -55 angstrom”: If I get it correctly, the value of -55 angstrom corresponds to the right panel of Fig. 4f obtained from the integrated Berry curvature in the energy interval from -40 meV to the Fermi energy, as explained in the next paragraph of the manuscript. The readers might be confused when they see this value for

the first time in this paragraph. I suggest the authors add a brief explanation here simply by adding, e.g., “(see the following discussions)” after “-55 angstrom”.

Reviewer #2 (Remarks to the Author):

In the manuscript by Lee and colleagues, they studied the origin of non-linear Hall effect (NLHE) in NbIrTe₄ by transport, angle-resolved photoemission spectroscopy (ARPES), and theoretical calculations. They fabricated a Hall bar device with a 15-nm thick flake (~12 layers) and observed NLHE in second-harmonic transverse voltage at 2 K. Moreover, they found temperature dependence of the second-harmonic transverse voltage which change its sign at 150 K. Based on the observation, they proposed the NLHE and its temperature dependence are originated from Berry curvature dipole and electronic structure changes as a function of temperature. To confirm this proposal, they carried out ARPES experiment and directly observed some representative momentum-energy cuts with their temperature dependence. From energy distribution curves (EDCs) at both the G and Y, they found slight shifts of the peaks in EDCs at -0.3 eV toward Fermi level (EF), thus argued chemical potential is shifting down as temperature increases. To support all the observation by Berry curvature dipole textures, they presented density functional theory calculations from bilayer and 12-layer cases. Berry curvature dipole along the a axis, D_a , is calculated with respect to energy. The calculation for the bilayer case confirmed that the dramatic change in D_a is due to the spin-orbit split bands that introduces a pair of Berry curvature monopoles. Extending this result, they calculated energy dependence of D_a in the 12-layer case, which further support their chemical potential shift scenario for the observed NLHE and temperature dependence.

The current work is similar to ref. 6 in the main text, but with detailed analyses to trace the microscopic origin of NLHE. It is important to understand novel transport properties not only by theoretical calculations and proposals but also by experimental observation, and this work has certain importance in this context. On the other hand, ARPES data quality does not seem better than a previous report, Ref. 27, and I have some major concerns whether ARPES can really support the proposed mechanism. Thus, I would like to hold my recommendation for publication until the points noted below are fully addressed.

1) The most critical observation for temperature dependence of chemical potential is Fig. 3c, meanwhile I am unconvinced by authors' presentation for temperature-dependent shifts in the current form. Authors' points make sense only if fittings for the entire EDC from EF to high enough binding energy are presented. In the current way of showing peak fittings for the partial range as shown in Fig. S4, I am unsure that the chemical potential shift is real. For example, the feature near

EF seems shifting to higher binding energy when temperature increases, which is the opposite direction.

2) Comparing their work with a previous ARPES report (ref. 27), I wonder whether they have fully examined surface-dependent features. If so, which surface did they observe and how did they cross-check it? In addition, referring the Figs. 2 and 3 in Ref. 27, the -0.3 eV band dispersion seems highly depending on surface. Could this surface dependence be relevant with temperature-dependent chemical potential shift?

3) I infer that another ARPES dataset at 10 K exist from line 160 in page 8 of the main text, but they did not present it. Can authors provide more dataset for transparent presentation? In addition, how did they fit EDCs at "C"? As I elaborated above, fittings should be made with large enough consideration for energy range.

4) In ref. 6, the temperature where NLHE sign flips in TaIrTe₄ is 175 K, which is quite similar to the case of NbIrTe₄. Considering Ta or Nb character near EF, spin-orbit splitting energy scale can be different in two cases, and if so, they should have different energy scale, thus resulting in different NLHE sign inversion temperatures. Can authors apply their scenario to TaIrTe₄ and comment the role of spin-orbit splitting energy scale?

5) They presented first-harmonic Hall voltage in Fig. S1. The description said that this data came from 20 nm flakes. Is this device different from the 15-nm flake device used to measure NLHE? If so, what is the reason they did not measure both on the same device?

6) How they confirm thicknesses of their devices?

7) A minor point- Calculated band structures by red dotted lines in Fig. 3b blocked experimentally obtained spectra. Can the authors have a better way of presentation so readers can access both the experimental spectra and comparison?

8) Another minor point to Fig. 3d- the grey ellipse with finite transparency alters the color of blue data points (EDCs at G). When I read the manuscript on another display device, the blue data points behind the grey ellipse looked like purple and confused me for a while. I would like to ask authors to consider a better way of presentation.

Reviewer #3 (Remarks to the Author):

Lee et al. measured the nonlinear Hall effect in the material NbIrTe₄, and they found the interesting sign changes in the Hall conductivity at 150K. They explain this phenomenon by the chemical potential shift at different temperatures. Moreover, they used angle resolved photoemission spectroscopy and density functional theory-based calculations to further prove their statement. In my opinion, the overall conclusion is reasonable, and parts of their experimental data can be well

explained by the DFT calculations. I would like to recommend its publication in Nature Communications. Before the final publication, I have some comments and suggestions:

1. Is there any reason or physical picture to explain that the chemical potential shifts down with increasing temperature in this material?
2. It seems that the authors use the parameter of bulk tight-binding models for the thicker slab calculations. However, the parameters of slab tight-binding models may be very different from its bulk counterparts, and they may change a lot depending on the thickness. I am wondering whether the authors compared the band structure from tight-binding models and these from DFT calculations for 12-layer slab.
3. As well known, the calculated band structures, especially these states around Fermi level, may change a lot depending on the methods used (e.g., LDA, PBE, or even Hybrid functional methods). I am wondering whether the authors verified that different methods could give qualitatively same conclusions.
4. The authors define integrated BC on line 236, while the math expression of the definition is confusing and misleading. BC can be a function of the band index n or the energy E but not both. In addition, it is not clear to me that why we need the integrated BC can represent the contribution to BCD.
5. The meaning of horizontal axis and vertical axis in Fig. 4f is not clear. They authors may add some labels.
6. Several previous works have shown the experimental and DFT results of NbIrTe₄ (e.g., Journal of the American Chemical Society, 114, 8963-8971 (1992); Nano letters 17, 467-475 (2017); Advanced Electronic Materials 5, 1900250 (2019)) . They authors may discuss whether the current results are consistent with previous related results.

Manuscript ID: NCOMMS-23-08582-T

Title: Spin-orbit-splitting-driven nonlinear Hall effect in NbIrTe₄

Authors: Ji-Eun Lee; Aifeng Wang; Shuzhang Chen; Minseong Kwon; Jinwoong Hwang; Minhyun Cho; Ki-Hoon Son; Dong-Soo Han; Jun Woo Choi; Young Duck Kim; Sung-Kwan Mo; Cedomir Petrovic; Choongyu Hwang; Se Young Park; Chaun Jang; Hyejin Ryu

We sincerely thank the reviewers for their suggestions and comments, which have been valuable for us to significantly improve the quality of our manuscript. Below are our point-by-point responses to the reviewers' comments. The reviewer's comments are shown in italics in the order appeared on the report, followed by our responses in sans-serif font. The manuscript has been revised following the advices of all the reviewers as summarized at the end of this document.

Reply to Reviewer #1

Comments:

The authors reported the observation of the nonlinear Hall effect (NLHE) in NbIrTe₄ thin flakes, attributed to the Berry curvature dipole (BCD) from the broken inversion symmetry while the time-reversal symmetry is preserved. Combining ARPES and DFT calculations, the authors concluded that the BCD is governed by the momentum texture of Berry curvature hotspots from partially occupied spin-orbit split bands, providing an explanation for the temperature dependence of the NLHE signal—especially the occurrence of a sign change.

The results presented in the manuscript represent a relatively thorough investigation of the NLHE in NbIrTe₄. However, I am not convinced how this work represents a significant step forward compared to the established literature on similar topics—see, e.g., Ref. 6 in the manuscript, which addresses the NLHE in TaIrTe₄. The authors simply stated that “Most of the studies on NLHE, however, have focused on transport properties and their applications.” From my understanding, the methodology adopted and the main physical implications in Ref. 6 are similar to that reported here in this work—except for a lack of ARPES evidence for the chemical potential shift with

temperature in Ref. 6. The energy- and momentum-resolved distribution of the BCD from calculations and the correlation with the NLHE have been discussed for TaIrTe₄. Therefore, the results presented in this manuscript for NbIrTe₄ certainly add to the existing knowledge about the NLHE but seem to be not of sufficient significance and novelty to be published in Nature Communications.

Our reply: We extend our sincere appreciation to the Reviewer#1 for recognizing the “thorough investigation” of the NLHE (nonlinear Hall effect), particularly using ARPES (Angle-resolved photoemission spectroscopy). Concurrently, we do agree with the Reviewer#1 for pointing out that our paper might not have sufficiently emphasized certain pivotal and novel aspects. In alignment with the reviewer’s insightful opinion, we have undertaken substantial revision to clearly state the significance and novelty of our work in the abstract, introduction, and Fig.1 part of our manuscript (page 2 lines 27-38, page 3 lines 58-59 and 64-71, and page 6 lines 111-114, respectively). Presented below is a summary of the novel findings uncovered in this study, which have been incorporated into and accentuated within the main manuscript.

Introduce the room-temperature non-linear Hall effect

First of all, we report a new family member of the NLHE material that exhibits NLHE even at room-temperature. Moreover, the Berry curvature dipole (BCD) is turned out to be the key factor not only in the realization of NLHE, but also in the manipulation of NLHE via the change of its sign, providing a viable route to tune NLHE using BCD, that has been rarely observed so far. Our finding will stimulate further efforts exploring the origin of this exotic phenomena and searching for additional family members, which will ignite a new paradigm of designing and manipulating quantum devices that can be operated beyond the quantum efficiency limit. This highlights not only novelty, but also significance of our work that will benefit general audience of Nature Communications.

Unveil the microscopic origin of exotic transport properties based on direct experimental and theoretical evidence

We also would like to emphasize the significant role of ARPES in understanding the

origin of the sign reversal of BCD in NbIrTe₄. As the reviewer already notices, BCD plays an important role in driving and tuning NLHE and hence an efficient way to tune BCD will invite huge interests in scientific as well as industrial fields. Despite it is trivial to expect that BCD can be controlled by the change of the electron band structure and several theoretical suggestions, it is interesting to note that a systematic study combining the NLHE and the experimental electron band structure that are interconnected by the BCD has not been reported yet as far as we notice, highlighting additional novelty of our work.

Furthermore, a previous work [Y. Jian et al., *2D Mater.* **8**, 015020 (2021)] on the chemical potential in TaIrTe₄ showed downward shift, i.e., hole doping, upon increasing temperature, in striking contrast to another work reported in Ref. 6, where the chemical potential exhibits upward shift, indicating electron doping. This controversy raises an important issue on the origin and the role of the chemical potential shift and hence invites experimental efforts to directly measure the electron band structure of a NLHE material, that we have done in our work, highlighting additional significance of our work.

We compare our findings and methodology with those of previous works in Table 1. Although numerous efforts have employed several different methods such as DFT and gating, an ARPES study that is the most powerful tool to directly provide information on the electron band structure has not been performed yet. Therefore, our work provides a missing link between BCD and NLHE via ARPES analysis combined with theoretical calculation.

Theoretical work elucidating the origin of the BCD

Finally, our calculations unveil a novel BCD "hot spot" emerging from electron bands that split by spin-orbit coupling. This is in contrast to the previous theoretical works that have claimed that the tilted band dispersion in Weyl or Dirac semimetals hosts the BCD. Our finding illuminates one of the plausible explanations on the origin of BCD.

Table 1. List of experimental reported NLHE.

Material	Temperature (K)	Origin	NLHE Sign reversal	Chemical potential study	Reference
WTe ₂ (bilayer)	10-100	BCD	x	Gating/DFT	Nature 565 , 17 (2019)
WTe ₂ (few layer)	1.8-100	BCD	x	x	Nat. Mater. 18 , 324-328 (2019)
Corrugated bilayer graphene	1.5-15	BCD	x	Gating/DFT	Nat. Electron. 4 , 116-125 (2021)
Twisted bilayer graphene	1.7-80	Skew	x	x	Phys. Rev. Lett. 129 , 186801 (2022)
MoTe ₂ (bulk)	2-40	Skew	x	x	Nat. Comm. 12 , 2049 (2021)
MoTe ₂ (bilayer)	10-100	BCD	x	DFT	Nat. Comm. 13 , 5465 (2022)
BaMnSb ₂	200-400	BCD	x	DFT	Nat. Comm. 14 , 364 (2023)
TaIrTe ₄	2-300	BCD	○	Hall/DFT	Nat. Nanotech. 16 , 421-425 (2021)
NbIrTe₄	2-300	BCD	○	ARPES/DFT	Our work

Minor comments:

1. Since the manuscript focuses on the NLHE, the completeness of the manuscript would benefit from the presentation of more detailed characterization of the NLHE: e.g., by showing the absence of the NLHE when the current is applied along directions other than the *a* axis, showing the absence of dependence on the frequency of the current (while maintaining the double-frequency condition), by performing scaling analysis, etc. These details would further corroborate the NLHE nature of the observed transport behavior, help to exclude measurement artifacts, and shed light on its origins. This is merely a suggestion.

Our reply: We appreciate the Reviewer #1 for highlighting the necessity for a more comprehensive characterization of the NLHE. Your valuable input will enhance the

completeness of the discussion on transport properties of our work. We have added plots of the driving AC current direction and frequency-dependent nonlinear Hall voltage in Figs. S2 and S3 of SI (as shown in Fig. R1 and R2).

Figure R1. Driving AC current direction-dependent nonlinear Hall voltage ($V^{2\omega}$) in NbIrTe₄. **a.** The in-plane crystal structure with a unit cell denoted by the black rectangle. The mirror axis is parallel to the b -axis. The AC current is applied along the a - and b -axes, labeled as $V_{a-bb}^{2\omega}$ and $V_{b-aa}^{2\omega}$. **b.** The second-harmonic transverse voltage depending on the driving current direction along two different crystal axes. The inset shows an optical microscopic image of the measured device. The scale bar is in 1 μm scale.

We measured second harmonic transverse Hall voltage upon changing current direction along two different crystal axes, i.e., a - and b -axes denoted in Fig. R1. Considering the crystal symmetry, only the driving current along the a -axis is expected to exhibit the NLHE signal. Evidently, the second-harmonic Hall voltage ($V_{a-bb}^{2\omega}$) with driving current along the a -axis surpasses the second-harmonic Hall voltage ($V_{b-aa}^{2\omega}$) with driving current along the b -axis, suggesting the absence of NLHE along the b -axis. We also measured the second harmonic transverse Hall voltage at several different frequencies of 117.77 Hz, 77.77 Hz, 37.77 Hz, and 17.77 Hz as shown in Fig. R2. We clearly observe

no frequency dependence in the measurements, suggesting that we can exclude measurement artifacts, as detailed on page 7 lines 131-133 in main text.

Figure R2. Nonlinear Hall voltage ($V^{2\omega}$) for several different frequencies of driving AC current in NbIrTe_4 . The second-harmonic voltage for different driving current frequencies of 117.77 Hz, 77.77 Hz, 37.77 Hz, and 17.77 Hz. The inset shows an optical microscopic image of the measured device with a 1 μm scale bar.

In our ongoing project, we are investigating the enhancement of the NLHE through the introduction of disorder-induced scattering in NbIrTe_4 . To comprehensively analyze the disorder-induced contributions, we are currently exploring the scaling law of NLHE. We intend to address this aspect in a separate paper, which will be published soon.

2. Line 231-232 of the manuscript: “which is similar to the theoretical value of -55 angstrom”: If I get it correctly, the value of -55 angstrom corresponds to the right panel of Fig. 4f obtained from the integrated Berry curvature in the energy interval from -40 meV to the Fermi energy, as explained in the next paragraph of the manuscript. The readers might be confused when they see this value for the first time in this paragraph. I suggest the authors add a brief explanation here simply by adding, e.g., “(see the following discussions)” after “ -55 angstrom”.

Our reply: We agree with the reviewer that readers can be confused when we use two different units without any explanation. Following reviewer's suggestion, we have revised our manuscript to add a brief explanation, as detailed on page 13 lines 251-253.

Reply to Reviewer #2

Comments:

In the manuscript by Lee and colleagues, they studied the origin of non-linear Hall effect (NLHE) in NbIrTe₄ by transport, angle-resolved photoemission spectroscopy (ARPES), and theoretical calculations. They fabricated a Hall bar device with a 15-nm thick flake (~12 layers) and observed NLHE in second-harmonic transverse voltage at 2 K. Moreover, they found temperature dependence of the second-harmonic transverse voltage which change its sign at 150 K. Based on the observation, they proposed the NLHE and its temperature dependence are originated from Berry curvature dipole and electronic structure changes as a function of temperature. To confirm this proposal, they carried out ARPES experiment and directly observed some representative momentum-energy cuts with their temperature dependence. From energy distribution curves (EDCs) at both the G and Y, they found slight shifts of the peaks in EDCs at -0.3 eV toward Fermi level (EF), thus argued chemical potential is shifting down as temperature increases. To support all the observation by Berry curvature dipole textures, they presented density functional theory calculations from bilayer and 12-layer cases. Berry curvature dipole along the a axis, D_a , is calculated with respect to energy. The calculation for the bilayer case confirmed that the dramatic change in D_a is due to the spin-orbit split bands that introduces a pair of Berry curvature monopoles. Extending this result, they calculated energy dependence of D_a in the 12-layer case, which further support their chemical potential shift scenario for the observed NLHE and temperature dependence.

The current work is similar to ref. 6 in the main text, but with detailed analyses to trace the microscopic origin of NLHE. It is important to understand novel transport properties not only by theoretical calculations and proposals but also by experimental observation, and this work has certain importance in this context. On the other hand, ARPES data quality does not seem better

than a previous report, Ref. 27, and I have some major concerns whether ARPES can really support the proposed mechanism. Thus, I would like to hold my recommendation for publication until the points noted below are fully addressed.

Our reply: We appreciate Reviewer #2's recognition of our manuscript, which presents comprehensive analyses of experimental observation (ARPES) and theoretical background (DFT) to unveil the microscopic origin of novel transport properties (NLHE). We also emphasize this point as a core argument in our paper. At the same time, we do agree with the reviewer's major concern that the current presentation of ARPES data may not be convincing enough to support our conclusion. To address this concern, we have made substantial revisions to our ARPES part as follows.

1) The most critical observation for temperature dependence of chemical potential is Fig. 3c, meanwhile I am unconvinced by authors' presentation for temperature-dependent shifts in the current form. Authors' points make sense only if fittings for the entire EDC from E_F to high enough binding energy are presented. In the current way of showing peak fittings for the partial range as shown in Fig. S4, I am unsure that the chemical potential shift is real. For example, the feature near E_F seems shifting to higher binding energy when temperature increases, which is the opposite direction.

Our reply: We appreciate the reviewer pointing out this crucial point, and we do agree with that current presentation could raise concerns on the analysis of ARPES data. To confirm the temperature-dependent chemical potential shift and to track the peak position more accurately, we have further analyzed our ARPES data using widely adopted "second derivative method".

In Figs. R3a and R3b, we show energy distribution curves (EDCs) and corresponding second derivative curves (SDCs) taken at 20 K, 25 K, 50 K, 70 K, 150 K, 180 K, and 240 K. The black dashed lines are guides for the eyes positioned at the peak at 20 K. A peak at $E-E_F \sim -0.3$ eV from the 20 K curve shifts towards lower binding energy in both EDC and SDC. Additionally, we display the second derivative ARPES cuts along the X- Γ -X

direction in the lower panels of Figs. R4, also showing peak shifts towards lower binding energy.

On the other hand, as the reviewer pointed out, the peak at $E-E_F \sim -0.05$ eV near the Fermi energy seems to show different behavior from the peak at $E-E_F \sim -0.3$ eV. However, it should be noted that the temperature-dependence of an energy spectrum near the Fermi energy is not straightforward to understand requiring careful analysis for the following reasons: (i) Typically, the Fermi-Dirac distribution hinders to define the exact binding energy of a state, especially when it is very close to the Fermi energy as is the case of ours. (ii) Thermal excitations make the analysis ever more non-trivial as not only the quasiparticle states are broadened, but also the cut-off energy range of the Fermi-Dirac distribution is also broadened. This thermal broadening becomes even more pronounced at higher temperature. To be more precise, when transitioning from 20 K to 70 K, the thermal broadening effect is relatively unaffected, indicating a subtle trend towards lower binding energies. However, as temperatures continue to rise, there is abrupt shifts towards higher binding energies (Considering that $4k_B T$ is approximately equal to 0.05 eV at 150 K). As a result, it is ambiguous to determine the exact binding energy of a state close to the Fermi energy, especially at different temperatures.

To address this concern, we have tried to **divide EDCs by the Fermi-Dirac distribution function** convoluted with an instrumental resolution to eliminate thermal broadening effect. Although it is subtle, we do not see electron doping, i.e., shift towards higher binding energy, but steady hole doping when the peak at about $E-E_F \sim -30$ meV from the 20 K curve is compared to the weak and broad peak at about $E-E_F \sim 0$ eV from the 240 K curve (Fig. R3c). As a result, the analysis of the peak at higher binding energy could minimize any controversy in terms of the energy shift of the electron band structure.

Based on the qualitative analysis of the results shown in Fig. R3(a)-(c), we present the temperature dependence of the peak position in Fig. R3(d), which is updated from the Fig. 3(d) in the previous version of our manuscript. We can clearly observe ΔE shifts as temperature increase from 10 K to 280 K, which suggests the chemical potential shift by about 15-20 meV towards the lower binding energy for all of our data sets. This result agrees with the previous transport results showing increasing hole carrier density with increasing temperature [Q. -G. Mu et al., npj Quantum Inf. **6**, 55 (2021), Y. Jian et al., *2D*

Mater. **8**, 015020 (2021)].

Through the careful analysis of our ARPES data following reviewer's comments, we believe that now we deliver more concise and conclusive discussion on the mechanism for NLHE that certainly improved overall quality of our paper. We have replaced Figs. 3(c) and (d) in the previous version of our manuscript with Figs. R3(a), (b), and (d), and revised the main text accordingly (page 9, lines 173-174). In addition, Fig. R3(c) and Fig. R4 were included in Fig. S7 and Fig. S6 of SI.

Figure R3. The change in the ARPES spectra for NbIrTe₄ with increasing temperature. a.

EDCs at the Γ point are shown for several different temperatures. The black dashed lines are guides for the eyes. **b.** The second derivative of the EDCs. **c.** EDCs divided by the Fermi-Dirac distribution function convoluted with an instrumental resolution. **d.** The energy shifts taken from two different spot in the momentum space (Γ and C). Hole carrier density (n) is obtained from the previous Hall measurements [Q. -G. Mu *et al.*, *npj Quantum Materials* **6**, 55 (2021)].

Figure R4. Temperature-dependence of ARPES intensity plots along the $\bar{X}-\bar{\Gamma}-\bar{X}$ direction at temperature of 20 K, 70 K, 180 K, and 240 K, respectively. Lower panels show corresponding second derivative plots. The white dashed line is aligned to the peak of the intensity around $E-E_F = \sim -0.3$ eV at 20 K, indicated by a red arrow.

2) Comparing their work with a previous ARPES report (ref. 27), I wonder whether they have fully examined surface-dependent features. If so, which surface did they observe and how did they cross-check it? In addition, referring the Figs. 2 and 3 in Ref. 27, the -0.3 eV band dispersion seems highly depending on surface. Could this surface dependence be relevant with temperature-dependent chemical potential shift?

Our reply: We agree with the reviewer that the surface-dependent feature may have an effect on our analysis of obtaining the temperature-dependent change of the chemical potential, given that surface-dependent features in the band structures around -0.3 eV in Ref. 27 [S. A. Ekahana et al., *Phys. Rev. B* **102**, 085126 (2020)]. We have examined the surface termination for the ARPES measurements based on the surface-dependent features of the Fermi surface. In NbIrTe₄, there are two different surface terminations, top and bottom, as shown in Fig. R5a, denoted by S1 and S2, respectively, following the definition of Ref. 27 [S. A. Ekahana et al., *Phys. Rev. B* **102**, 085126 (2020)]. We measured Fermi surfaces for three different samples as shown in Fig. R5b and found that all the samples show elliptical hole pockets at the Y point. To identify surface-dependent features in the ARPES spectra, we performed termination dependent band structure calculations (panel **c**) that can be considered simulated ARPES measurements and found that only the S1 terminate surface has elliptical hole pockets at Y point, consistent with the Fermi surface in Ref 27 [S. A. Ekahana et al., *Phys. Rev. B* **102**, 085126 (2020)]. Thus, the comparison between measured and simulated ARPES spectra suggests that the Fermi structure features show the characteristics of band structures with S1 termination.

In addition to identifying the surface termination, we checked whether there are surface-dependent changes in the band structures induced by the chemical potential shift using theoretical calculations. The panels (**d-i**) in Fig. R5 present the bulk and surface band structures for undoped and hole-doped NbIrTe₄, in which the hole-doping effect is included by reducing the number of electrons (0.025 h/f.u.) with a compensating uniform background charge. The amount of doping is chosen such that the chemical potential shift of the bulk band structures is about 20 meV, close to the estimated value from ARPES (panels **d** and **e**). We find that the surface band structures along the Γ -X high-symmetry line for both S1 and S2 terminations exhibit uniform upward shifts with amounts almost identical to the bulk bands. Thus, the temperature-dependent shift around -0.3 eV that we identify with ARPES spectra can be interpreted as the approximately same chemical potential shift, regardless of the surface termination.

In response to the reviewer's question, we revised the main text accordingly (page 10, lines 178) and we have also included Fig. R5 in Fig. S13 of SI.

Figure R5. Surface termination dependence of electronic band structures of NbIrTe₄. **a.** Schematics showing top and bottom surfaces, denoted as S1 and S2, respectively. **b.** The Fermi surfaces obtained for three independent experiments (i)-(iii) using different samples, measured at 20 K, 20 K, and 10 K, respectively. The blue-dashed boxes show double hole pockets near the Y point that exist only on the S1 terminated surface. **c.** Fermi surfaces of surface band structures

calculated for S1 (top panel) and S2 termination (bottom panel). Bulk band structures along Γ -X high symmetry line for pristine (**d**) and 0.025 h/f.u. doped (**e**) cases. The solid black and dashed magenta lines denote the Fermi energy and the binding energy of -0.3 eV, respectively. **f-i**. Surface band structures along Γ -X high symmetric direction: (**f**) pristine case with S1 termination, (**g**) 0.025 h/f.u. doped case with S1 termination, (**h**) pristine case with S2 termination, (**i**) 0.025 h/f.u. doped case with S2 termination. The bottom panels of (**d-i**) show band structures around the -0.3 eV for better visualization of the vicinity.

3) I infer that another ARPES dataset at 10 K exist from line 160 in page 8 of the main text, but they did not present it. Can authors provide more dataset for transparent presentation? In addition, how did they fit EDCs at “C”? As I elaborated above, fittings should be made with large enough consideration for energy range.

Our reply: Regarding the EDC fit at the C point, we provide an example in Fig. R6a, where we show ARPES data taken at 10 K (what the reviewer requested) and indicate the C point (0.13 \AA^{-1}) with the green-dashed line. Figure 6b shows wide range EDCs taken at the C point.

To address the reviewer’s comment, we have performed wide range fitting. Figure R7a shows the three-peak fit to the 10 K and 280 K EDCs taken at the C point that we showed in the old version of our manuscript. The black-solid line is a fit result using a Lorentzian multi-peak function for the energy range from -0.04 eV to -0.56 eV. Figure R7b shows our new fit to the EDCs for the wide energy range from -0.02 eV to -1.43 eV, including linear background, denoted by the grey-dashed line. Figure R7c is the summary of our fit focused on the -0.3 eV peak (denoted by the green arrow) for both short and long energy ranges. We found no substantial difference between the two results. We have revised the supplementary information to discuss our fit in Figs. S8 and S9.

Figure R6. ARPES data taken at 10 K and EDCs taken at the C point. **a.** ARPES intensity map at $E-E_F = 0$ eV and cuts taken along the $\bar{\Gamma}$ - \bar{X} and \bar{Y} - \bar{S} direction at 10 K. The green-dashed line positioned at 0.13 \AA^{-1} denotes the C point. **b.** Temperature-dependent EDCs taken from the C point. The green arrow denotes a fitted peak position, while the black-dashed line is guide to the eyes.

Figure R7. A comparison of multi-peak fitting for different energy ranges. a. Multi-peak fit to 10 K and 280 K EDCs for an energy range from -0.04 eV to -0.56 eV. **b.** Multi-peak fit to EDCs to 10 K and 280 K EDCs for a wider energy range from -0.02 eV to -1.43 eV. Lorentzian peak function has been used for all the fit. In the short-range fit, background has been considered with two peaks (orange- and blue-dashed curves) above and below the -0.3 eV peak, whereas the wide-range fit employed five peaks after subtracting linear background. **c.** Comparison of two fit results by showing energy shifts ($\Delta E = E - E_{T=10\text{K}}$) as a function of temperature.

4) In ref. 6, the temperature where NLHE sign flips in TaIrTe₄ is 175 K, which is quite similar to the case of NbIrTe₄. Considering Ta or Nb character near EF, spin-orbit splitting energy scale can be different in two cases, and if so, they should have different energy scale, thus resulting in different NLHE sign inversion temperatures. Can authors apply their scenario to TaIrTe₄ and comment the role of spin-orbit splitting energy scale?

Our reply: We thank the reviewer for bringing this important issue. The potential influence of spin-orbit splitting on both the BCD and NLHE response could be an important consideration, as supported by relevant references [Z. Z. Du et al., *Phys. Rev. Lett.* **121**, 266601 (2018), Q. Ma et al., *Nature* **565**, 337-342 (2019)]. As pointed out by the reviewer, tantalum (Ta) possesses larger spin-orbit splitting energy scale than that of niobium (Nb). Consequently, our observations of NLHE in NbIrTe₄ exhibits a transition temperature and magnitude of NLHE, which are relatively smaller than those observed in TaIrTe₄ [ref.6]. We have revised the main text to discuss this interesting issue, supporting our scenario (page 6 lines 137-140).

5) They presented first-harmonic Hall voltage in Fig. S1. The description said that this data came from 20 nm flakes. Is this device different from the 15-nm flake device used to measure NLHE? If so, what is the reason they did not measure both on the same device?

Our reply: First of all, transport measurements demand meticulous attention due to sample's vulnerability to breakage. Even minor static electric shocks can result in device damage, even during routine electrode manipulation. Unfortunately, despite of our careful handling, the 15 nm-thick device suffered damage, preventing us from measuring the first-harmonic response. Alternatively, we decided to measure another sample, and with this replacement, we successfully obtained the first-harmonic response.

To ensure the reliability of our first-harmonic data, we conducted multiple experiments. Figure R8 displays the first-harmonic responses of three samples, labeled as Sample 1, Sample 2, and Sample 3, respectively, with two experimental geometries illustrated as green bars. While $V_{||}^{\omega}$ (longitudinal) and V_{\perp}^{ω} (transverse) measurements in all three

samples exhibit a linear response to applied current, the values of V_{\perp}^{ω} are significantly smaller than those of V_{\parallel}^{ω} . This observation indicates a consistent absence of the linear Hall response (V_{\perp}^{ω}) due to the preserved time-reversal symmetry.

We recognize that the readers may have similar concerns regarding first-harmonic response. As a result, we have revised our figure and its corresponding explanation in Figure S1 of SI.

Figure R8. First-harmonic V_{\parallel}^{ω} (longitudinal) and V_{\perp}^{ω} (transverse) as function of the AC current I^{ω} for three different samples at 2 K. (Sample 1, Sample 2, and Sample 3). All measured V_{\parallel}^{ω} values (closed symbols) consistently exceed the corresponding V_{\perp}^{ω} values (open symbols), indicating the disappearance of the linear Hall response (V_{\perp}^{ω}) due to preserved time-reversal symmetry. The green crossbars illustrate the experimental geometries. V_{\parallel}^{ω} and V_{\perp}^{ω} represent the first-harmonic voltage parallel and perpendicular to the applied current direction (four probe I/V measurement and Hall measurement), respectively. The applied current is injected from the source (S) electrode to the drain (D) and the voltage is measured between A and B electrodes.

6) How they confirm thicknesses of their devices?

Our reply: We confirmed the thickness of an exfoliated NbIrTe₄ using atomic force microscopy (AFM). The AFM image in Fig. R9a shows a 15 nm-thick NbIrTe₄ flake on SiO₂ substrate. The height profile of the NbIrTe₄ flake in Fig. R9b was taken along the red line in panel a. We have included AFM data in Fig. S5 of SI.

Figure R9. Determination of the thickness of an exfoliated NbIrTe₄ flake by AFM. a. An AFM image showing a 15 nm-thick flake of NbIrTe₄ on an SiO₂ substrate. b. The measured height profile along the red line in panel a.

7) A minor point- Calculated band structures by red dotted lines in Fig. 3b blocked experimentally obtained spectra. Can the authors have a better way of presentation so readers can access both the experimental spectra and comparison?

Our reply: We appreciate the reviewer for pointing this out. In response to the reviewer's comment, we have modified Fig. 3b to show the ARPES data. Hopefully, we believe that Figure 3b now have a better presentation, allowing for a more comprehensive comparison with the experimental spectra and calculations.

8) Another minor point to Fig. 3d- the grey ellipse with finite transparency alters the color of blue data points (EDCs at G). When I read the manuscript on another display device, the blue data points behind the grey ellipse looked like purple and confused me for a while. I would like to ask authors to consider a better way of presentation.

Our reply: I agree with the reviewer's suggestion. Following reviewer's comment, we have included the yellow ellipse in the background in Fig. 3d. I hope this will enhance the visual presentation.

Reviewer #3 (Remarks to the Author)

Lee et al. measured the nonlinear Hall effect in the material NbIrTe₄, and they found the interesting sign changes in the Hall conductivity at 150K. They explain this phenomenon by the chemical potential shift at different temperatures. Moreover, they used angle resolved photoemission spectroscopy and density functional theory-based calculations to further prove their statement. In my opinion, the overall conclusion is reasonable, and parts of their experimental data can be well explained by the DFT calculations. I would like to recommend its publication in Nature Communications. Before the final publication, I have some comments and suggestions:

Our reply: We thank the reviewer for recommending our manuscript for publication and for valuable comments. We have fully responded to the reviewer's comments and suggestions as shown below. We believe our manuscript has become clearer and more convincing thanks to the fruitful comments and suggestions of the reviewer.

1. Is there any reason or physical picture to explain that the chemical potential shifts down with increasing temperature in this material?

Our reply: We thank the reviewer for the insightful question. There could be several possible mechanisms that lead to the chemical potential shift such as the charge carrier localization/delocalization by temperature [Colinge J-P and Colinge C A 2005 Physics of

Semiconductor Devices (New York, USA: Springer Science & Business Media)], electron-phonon couplings [Phys. Rev. B 96, 100504(R)], and thermal expansion [Phys. Rev. Lett. 124, 236601 (2020), Phys. Rev. Lett. 130, 236402 (2023)]. However, it is indeed challenging to identify the dominant factor.

We find that our temperature-dependent Raman data shows a redshift with increasing temperature, shown in Fig. R10, suggesting the volume expansion, which may lead to the chemical potential shift. However, we believe that finding the volume expansion only may not be enough to rule out other possible mechanisms, especially without the information on the temperature-dependent lattice parameters and associated change in the atomic positions [Phys. Rev. Lett. 110, 067002 (2013)]. While we do not have a clear answer, finding the physical mechanism for the chemical potential shift calls for further investigation and is our subject for future study.

Figure R10. Temperature-dependent Raman shift for NbIrTe₄. Temperature-dependent Raman spectra for NbIrTe₄ at several different temperatures. Four peaks correspond to the

Raman mode (A1 and B1) characterized in the reference [H. Chen et al., *Solid State Commun.* **289**, 56-60 (2019)].

2. It seems that the authors use the parameter of bulk tight-binding models for the thicker slab calculations. However, the parameters of slab tight-binding models may be very different from its bulk counterparts, and they may change a lot depending on the thickness. I am wondering whether the authors compared the band structure from tight-binding models and these from DFT calculations for 12-layer slab.

Our reply: We thank the referee for the question. We agree with the referee that the use of the bulk tight-binding parameters in calculating the slab geometry needs to be justified. Following the referee's suggestion, the band structures of the supercell tight-binding Hamiltonian are compared with the corresponding DFT results for vacuum-slab geometry of two-, four-, and six-layer slabs, showing almost identical band structures. We note that the DFT calculations of the thicker slabs with the inclusion of spin-orbit coupling exhibit poor convergence, which is the main reason why the supercell tight-binding Hamiltonians are used to calculate the Berry curvature dipole of the thick slab. Based on the excellent agreement in the electronic structure up to the six-layer slab, we believe that the bulk tight-binding parameters can be used in the slab calculations for our system.

Following the referee's suggestion, we have added a sentence, "The slab band structures of the supercell tight-binding Hamiltonian and those of DFT using vacuum-slab geometries are in good agreement (Fig. S11 in SI)." to the Methods section of the manuscript (page 16, lines 321-333) and added Fig. S11 (Fig. R10) to SI.

Figure R11. Comparison of slab band structures between the supercell tight-binding (TB) and DFT calculations. a-c, The TB band structures for two, four, and six-layers. d-f, The DFT band structures for two, four, and six-layer slabs.

3. As well known, the calculated band structures, especially these states around Fermi level, may change a lot depending on the methods used (e.g., LDA, PBE, or even Hybrid functional methods). I am wondering whether the authors verified that different methods could give qualitatively same conclusions.

Our reply: We agree with the reviewer that different exchange-correlation potentials may lead to qualitatively different electronic structures. As the referee suggested, we calculate the band structures using the LDA and HSE06 hybrid functional and compare the electronic structures and spatial texture of the Berry curvature. We compare the band structures in the bilayer slab since the system is mainly analyzed to reveal the characteristics of the band structures (spin-orbit-induced band splitting) that dominantly

contribute to BCD, and practically, performing the Hybrid functional calculations in a large system with the inclusion of spin-orbit requires high calculation cost.

With the LDA exchange-correlation potential, we find almost identical band structures, Berry curvature dipole, and momentum-dependent Berry curvature texture, leading to the same conclusion that the SOC-induced splitting dominantly contributes to the Berry curvature dipole, as presented in Fig. R12.

With the HSE06 hybrid functional, we find the ground state with stripe-type ferrimagnetic ordering with a small Nb local magnetic moment of $0.08 \mu_B$ but with negligible net magnetization, as shown in Fig. R13. In our collinear HSE06 calculations, we find that the ferrimagnetic phase is about 2 meV/f.u. lower in energy than the non-magnetic phase and also find that setting a ferromagnetic ordering is unstable and relaxes to ferrimagnetic ordering. With spin-orbit coupling, the magnetic moment is in the bc -plane with moment dominantly along the c -axis, breaking the M_a mirror symmetry and time reversal symmetry. As a result, the degeneracy of the Γ -point is lifted, as shown in the red dotted circle in the band structures. Since there is no report of the magnetism of NbIrTe₄ thin films, we believe that the use of the hybrid functional, at least for the bilayer slab, is not relevant.

Following the reviewer's comment, we have added a sentence "We note that the BCD and momentum-resolved BC of the bilayer slab are insensitive to the choice of the local density approximation (LDA) or generalized gradient approximation (GGA) for the exchange-correlation potential (see Fig. S12 in SI)." In page 12, lines 238-241 of the revised manuscript. We have added Fig. S12 (Fig, R12) to SI and the information on the LDA exchange-correlation functional to the Method section.

Figure R12. Comparison of BCD and momentum-resolved BC of the bilayer slab between LDA and GGA exchange-correlation functionals. a-b. Calculated band structures along the high-symmetry lines using the GGA (panel a) and LDA (panel b) exchange-correlation functionals. c. The BCD (D_a) calculated within the LDA for bilayer NbIrTe₄ plotted as a function of the chemical potential. d. Momentum-resolved BC ($\Omega_c(\mathbf{k})$ in \AA^2) calculated within the LDA for bilayer NbIrTe₄ and integrated from -11 to -6 meV corresponding to the red shaded area in panel c. e. Band dispersion of bilayer NbIrTe₄ around the k -points (top panel), where the contribution to the Berry curvature (lower panel) is large. The black-dashed lines in the upper panel are the bands calculated without spin-orbit coupling. The lower panel is BC ($\Omega_c(\mathbf{k})$) integrated up to -7.5 meV with respect to the Fermi energy (E_F , red dashed line).

Figure R13. Magnetic ordering and band structures calculated by HSE06 hybrid functional.
a. Magnetic ordering pattern of the ground state of the bilayer slab, where Nb1-Nb4 denotes inequivalent Nb sites. **b.** Band structures along the high-symmetry lines. The red dashed circle denotes the lifted degeneracy at the Γ -point.

4. The authors define integrated BC on line 236, while the math expression of the definition is confusing and misleading. BC can be a function of the band index n or the energy E but not both. In addition, it is not clear to me that why we need the integrated BC can represent the contribution to BCD.

Our reply: We thank the reviewer for pointing out the misleading expression for the integrated BC and agree with the reviewer that BC cannot be a function of both energy and a band index at the same time. We have corrected the equation as $\Xi_k(E) = \sum_n \int_{E_F-40 \text{ meV}}^{E_F+E} dE' \delta(E' - \epsilon_{nk}) \Omega_{nk,c}$. The BCD along the a -axis $D_a(E)$ is proportional to $\frac{\partial \Omega_{nk,c}}{\partial k_a}$ summed over the bands below the energy E . Therefore, when summed over a relatively narrow energy range, large BCD is expected around the k -points showing an abrupt change in $\Xi_k(E)$.

Following the reviewer's comment, we have added a sentence "Since the BCD along the

a -axis $D_a(E)$ is proportional to $\frac{\partial \Omega_{nk,c}}{\partial k_a}$ summed over the bands below the energy E . large

BCD is expected around the k -points showing an abrupt change in $\Xi_k(E)$ when integrated over a narrow energy range" in page 13, lines 258-261.

5. *The meaning of horizontal axis and vertical axis in Fig. 4f is not clear. They authors may add some labels.*

Our reply: We thank the reviewer for careful reading of our manuscript. We have modified Fig. 4f by adding labels and tick marks.

6. *Several previous works have shown the experimental and DFT results of NbIrTe₄ (e.g., Journal of the American Chemical Society, 114, 8963-8971 (1992); Nano letters 17, 467-475 (2017); Advanced Electronic Materials 5, 1900250 (2019)). They authors may discuss whether the current results are consistent with previous related results.*

Our reply: We thank the reviewer for providing relevant references and insightful suggestions. The suggested reference [Advanced Electronic Materials **5**, 1900250 (2019)], the experimental Hall measurements as a function of temperature exhibit notable consistency with our temperature-dependent ARPES results. It shows a significant increase in hole carriers with rising temperature which supports our argument. Furthermore, the DFT results in Nano letters **17**, 467-475 (2017), Advanced Electronic Materials **5**, 1900250 (2019), and American Chemical Society, **114**, 8963-8971 (1992), concerning bulk NbIrTe₄ are overall consistent with our calculation, further enhancing the reliability of our analysis. We have added these suggested references into the main text, specifically in the section where we compare our findings with previously published work, as detailed on page 9 line 168 and page 10 line 181. This inclusion has significantly bolstered the manuscript and advanced a more comprehensive understanding of our research.

List of changes

Changes in the manuscript

1. In the Manuscript page 1 lines 27-38, the sentence “The Berry curvature dipole (BCD)... transition-metal chalcogen compounds.” has been modified.

2. In the Manuscript page 3 lines 58-59, the sentence “characterized by a...AC driving current.” has been revised.
3. In the Manuscript page 3 lines 64-71, the sentence “in which the...of the BCD.” has been revised.
4. In the Manuscript, the Fig. 1e has been changed to new figure along with its captions.
5. In the Manuscript page 5 line 85, the caption title of Fig. 1 “**Crystal structure symmetry and band model of NbIrTe₄**” has been revised.
6. In the Manuscript page 5 line 97, the subtitle “**Prediction of the nonlinear Hall effect in NbIrTe₄**” has been revised.
7. In the Manuscript page 5 lines 107-109, the sentence “The devices were...Raman spectroscopy (Fig. 1d).” has been revised.
8. In the Manuscript page 6 lines 111-114, the sentence “In this system...subsequent theoretical section.” has been revised.
9. In the Manuscript page 7 lines 131-133, the sentence “Additionally, we have...experimental measurement artifacts.” has been revised.
10. In the Manuscript page 7 lines 137-140, the sentence “However, the slight...from Nb to Ta.” has been revised.
11. In the Manuscript page 8 lines 143-145, the title of Fig.3 “**Electronic structures and evidence of the temperature-dependent chemical potential shift of NbIrTe₄**” has been revised.
12. In the Manuscript, the Fig. 3a has been changed to new figure along with its caption.
13. In the Manuscript, Fig. 3c has been modified to have a better presentation, allowing for a more comprehensive comparison with the ARPES experiments and calculations.
14. In the Manuscript Fig. 3d and 3e have been changed to new figure (Fig. R3 a, b, and d) along with revised captions to provide a better presentation..
15. In the Manuscript page 9 lines 173-174, the sentence “Second derivative EDCs...In Fig. 3e” has been revised.
16. In the Manuscript page 10 lines 178, the phrase “which is consistent with our....(See Fig. S13 in SI).” has been revised.
17. In the Manuscript page 12 lines 238-241, the sentence “We note that...(see Fig. S12 in SI).” has been revised.

18. In the Manuscript page 12 lines 245, “(see Fig. S10 in SI)” has been revised.
19. In the Manuscript page 13 lines 251-252, “(see a section of...in SI)” has been revised.
20. In the Manuscript page 13 lines 252-253, the sentence “as explained in the following discussion” has been revised. In the Manuscript page 13 lines 258-261, the sentence “ $\Xi_k(E) = \sum_n \int_{E_F-40 \text{ meV}}^{E_F+E} dE' \delta(E' - \epsilon_{nk}) \Omega_{nk,c}$...a narrow energy range.” Has been revised.
21. In the Manuscript page 14 line 281, “that persists up” has been revised.
22. In the Manuscript page 14 lines 286-287, the sentence “and sign-changing BC...the slab calculation” has been revised.
23. In the Manuscript page 14, line 293, the sentence “from a single crystal...and transferred” has been revised.
24. In the Manuscript page 15 lines 310-312, the sentence “The Ceperley-Alder...(GGA), respectively” has been revised.
25. In the Manuscript page 16 lines 321-323, the sentence “The slab band structures...in good agreement (Fig. S6 in SI).” Has been revised.
26. In the Manuscript page 16 lines 325-326, “the surface bands were...WannierTools code.” has been revised.
27. In the Manuscript page 16 line 328-337, the supplementary information section has been revised including new supplementary contents.
28. In the Manuscript page 16 line 344, the acknowledgment of 2021R1A6A1A10044154 and 2022M3H4A1A04074153 has been added.
29. In the Manuscript page 16 lines 347-348, the sentence “J.-E. L. was...collaborative Postdoctoral Fellowship.” has been added in acknowledgment part.

Changes in the Supplementary Information

1. In the Supplementary Information, a section titled “**First harmonic Hall voltage of NbIrTe₄**” has been revised, along with new Fig. S1 and its caption.
2. In the Supplementary Information, a whole new section titled “**The direction and frequency of driving AC current-dependent second harmonic Hall voltage of NbIrTe₄**” along with a new figure S2 and S3 have been added.
3. In the Supplementary Information, a whole new section titled “**Determination of the**

thickness of the exfoliated NbIrTe₄ flake by atomic force microscopy (AFM)” along with a new figure of Fig S5 and its caption has been added.

4. In the Supplementary Information, a whole new section titled “**Temperature-dependent electronic band structure of NbIrTe₄**” along with new figures of Fig S6, S7, S8, and S9 have been added.
5. In the Supplementary Information, a whole new section titled “**Comparison of slab band structure of supercell tight-binding and DFT calculation**” along with a new figure of Fig. S11 and its caption has been added.
6. In the Supplementary Information, a whole new section titled “**Comparison of the band structures, BCD and momentum-resolved BC between the GGA and LDA schemes**” along with a new figure of Fig. S12 and its caption has been added.
7. In the Supplementary Information, a whole new section titled “**Determination of the surface termination and doping-dependent surface band structures**” along with a new figure of Fig. S13 and its caption has been added.

REVIEWER COMMENTS

Reviewer #1 (Remarks to the Author):

I appreciate the authors' detailed response to the reports. I believe my concerns have all been addressed. I can recommend publication in Nature Communications now.

A minor point: On page 9 of the response letter (in the reply to Reviewer #2), the authors stated "We can clearly observe ΔE shifts as temperature increase from 10 K to 280 K, which suggests the chemical potential shift by about 15-20 meV towards the lower binding energy for all of our data sets." If I get it correctly, the "lower" here should be a typo?

Reviewer #2 (Remarks to the Author):

I see that the authors tried to address questions by reviewers and improve their manuscript following the suggestions. However, I maintain my previous stance that the angle-resolved photoemission spectroscopy (ARPES) results presented in the manuscript are not convincing enough to support their argument. I had previously raised related concerns in the last report, and the authors' responses do not directly resolve them. As long as concerns regarding ARPES remain unresolved, I cannot recommend this work to be published in Nature Communications as other parts of this work (transport and theory) are simply applications of existing reports.

1) In the last report, I mentioned "Authors' points make sense only if fittings for the entire energy distribution curves (EDCs) from EF to high enough binding energy are presented". The authors showed second derivative curves and EDCs divided by Fermi-Dirac distribution (FD) instead of presenting fittings for EDCs. They suggested that thermal broadening of spectral intensity as well as that of the FD cut-off energy may explain why reliability cannot be guaranteed. However, I believe that raw EDC fittings with convoluted Lorentzian curves could work fine. Moreover, peak positions in EDCs and those divided by FD are deviating from each other, which questions whether the second derivative plot can accurately represent peak positions in raw EDCs. Lastly, it is doubtful that an energy resolution of 20 meV can detect such delicate changes as shown in Fig. 3d, thus additional high-resolution experiment is desired.

2) To investigate surface dependence, the authors provided three different Fermi surfaces together with theoretical calculations. They claimed that a small hole pocket at the Y point only exists on the S1 surface, therefore, all the displayed Fermi surfaces were expected to be from the same surface.

However, to me, it is unclear whether they observed the same surface. I do not see clear periodicity along the k_y axis as marked in the figure, and I am even unconvinced if the “pockets” are really at the Y point. Additionally, all three plots seem very different from each other, and even the second one more closely resembles Fermi surface from S2 (see Fig. 3(c) in Ref. 29 of the main text). As non-linear Hall effect is from surface band, it is very crucial to unambiguously determine surface termination. For the theoretical parts, I agree that -0.3 eV feature similarly shifts as a function of carrier concentrations. However, it is from bulk which is not expected to contribute Berry curvature dipole. Can they apply the argument to surface states (i.e. states lack of glide mirror symmetry) near the Fermi level?

3) Fittings for EDC at the C point (Fig. R7) look ambiguous as they obviously deviate near the Fermi level. This should be improved by convoluting FD and experimental resolution with Lorentzian curves. At this moment, both the short-range and long-range fittings provide no information. For instance, I can randomly set a “cut-off” energy where fitting results (black curve) start departing from experimental observation. This ultimately alters peak positions, especially peaks near the Fermi level and -0.3 eV, found from the fittings. Moreover, the C point is where a band dispersion crosses the Fermi level (Fermi momentum), but in the fittings I do not see any quasiparticle peak at the Fermi level even at 10 K.

Reviewer #3 (Remarks to the Author):

The authors have provided more benchmark calculations and addressed all my comments. I recommend its publication, while there are some loose ends to fix in the manuscript. For example, the related reference, Nano letters 17, 467-475 (2017) that predicted the existence of Weyl points the momentum-dependent Berry curvatures in these materials, is not added in the manuscript as said in the rebuttal letter.

Manuscript ID: NCOMMS-23-08582A

Title: Spin-orbit-splitting-driven nonlinear Hall effect in NbIrTe₄

Authors: Ji-Eun Lee; Aifeng Wang; Shuzhang Chen; Minseong Kwon; Jinwoong Hwang; Minhyun Cho; Ki-Hoon Son; Dong-Soo Han; Jun Woo Choi; Young Duck Kim; Sung-Kwan Mo; Cedomir Petrovic; Choongyu Hwang; Se Young Park; Chaun Jang; Hyejin Ryu

We sincerely thank the reviewers for their recommendation, suggestions, and comments, which have been valuable for us to significantly improve the quality of our manuscript. Below are our point-by-point responses to the reviewers' comments. The reviewer's comments are shown in italics in the order appeared on the report, followed by our responses in Arial font. The manuscript has been revised following the advice of all the reviewers as summarized at the end of this document.

Reply to Reviewer #1

Comments:

I appreciate the authors' detailed response to the reports. I believe my concerns have all been addressed. I can recommend publication in Nature Communications now.

Our reply: We deeply appreciate for the reviewer's acknowledgment of the novelty and significance of our work. We extend our sincere thanks for the positive decision and once again express our appreciation.

A minor point: On page 9 of the response letter (in the reply to Reviewer #2), the authors stated "We can clearly observe ΔE shifts as temperature increase from 10 K to 280 K, which suggests the chemical potential shift by about 15-20 meV towards the lower binding energy for all of our data sets." If I get it correctly, the "lower" here should be a typo?

Our reply: We agree with the Reviewer's comment. We acknowledge that "lower" binding energy was a typo in the previous response letter. The chemical potential shifts towards the "higher" binding energy when the electronic band (ARPES spectra) undergoes towards "lower" binding energy. We appreciate for your thorough and precise correction. We also double-checked our manuscript, and we would like to clarify that it accurately represents chemical potential shift towards "higher" binding energies. We deeply appreciate the reviewer's comment in bringing this to our attention.

Reply to Reviewer #2

Comments:

I see that the authors tried to address questions by reviewers and improve their manuscript following the suggestions. However, I maintain my previous stance that the angle-resolved photoemission spectroscopy (ARPES) results presented in the manuscript are not convincing enough to support their argument. I had previously raised related concerns in the last report, and the authors' responses do not directly resolve them. As long as concerns regarding ARPES remain unresolved, I cannot recommend this work to be published in Nature Communications as other parts of this work (transport and theory) are simply applications of existing reports.

Our reply: We appreciate the reviewer's recognition of our efforts to *address* reviewer's questions and *improve* the manuscript. We also acknowledge the reviewer's concerns regarding the necessity for a more definitive analysis of ARPES results, and we recognize that the previous version of reply letter did not fully address those concerns. In response, we have thoroughly examined and understood the specific concerns raised by the reviewer. Subsequently, we have taken comprehensive steps to ensure a more precise and definitive analysis of the data by following the reviewer's suggestions. We highly value the reviewer's comment, and we believe that our revised analysis makes our manuscript convincing enough.

1) In the last report, I mentioned “Authors’ points make sense only if fittings for the entire energy distribution curves (EDCs) from E_F to high enough binding energy are presented”. The authors showed second derivative curves and EDCs divided by Fermi-Dirac distribution (FD) instead of presenting fittings for EDCs. They suggested that thermal broadening of spectral intensity as well as that of the FD cut-off energy may explain why reliability cannot be guaranteed. However, I believe that raw EDC fittings with convoluted Lorentzian curves could work fine. Moreover, peak positions in EDCs and those divided by FD are deviating from each other, which questions whether the second derivative plot can accurately represent peak positions in raw EDCs. Lastly, it is doubtful that an energy resolution of 20 meV can detect such delicate changes as shown in Fig. 3d, thus additional high-resolution experiment is desired.

Our reply: We sincerely appreciate and agree with the insightful comments provided by the reviewer, particularly highlighting the importance of multiplying the Fermi-Dirac distribution (FD) function and convoluting instrumental resolution, ranging from the Fermi energy (E_F) to higher binding energies, as also emphasized in question #3. In response to these valuable suggestions, we have thoroughly conducted a newly **refined fitting with convoluted Lorentzian curves (multi-peak fit approach)** to comprehensively address and resolve concerns.

Our revised fitting analysis involves convoluting FD and experimental resolution with several Lorentzian curves to cover a wide range of energy at both the Γ and C point. Now, the fitted curves are accurately aligned with the Fermi level cut-off energy, as presented in Figure R1. Furthermore, when the temperature increases to 240 K (Γ point) and 280 K (C point), distinct peaks labeled as Peaks 1 and 2 (Γ point) and Peak (C point) are clearly resolved. We note that other peaks denoted with grey dotted lines in Figure R1 exhibit significant broadening at higher temperatures, displaying fluctuating weight during the fitting process, rendering them unreliable.

Figure R1. A modified multi-peak fitting of energy distribution curves (EDCs) at Γ and C point, which is fitted with Lorentzian curves, incorporating the multiplication of the Fermi-Dirac distribution (FD) function and convolution with instrumental resolution. (a) Multi-peak fit to 20 K and 240 K EDCs at the Γ point for energy range from -1.77 eV to Fermi energy (E_F). (b) Multi-peak fit to 10 K and 280 K EDCs at the C point for energy range from -1.43 eV to 0.09 eV.

Figure R2. The temperature-dependent energy shifts for the Γ and C points. The temperature-dependent energy shifts for the peak 1 and 2 of EDCs at the Γ point ($\Delta E = E - E_{T=20\text{ K}}$) and C point ($\Delta E = E - E_{T=10\text{ K}}$). n represents charge carrier density as function of temperature from ref. 28 through Hall measurement [Q. -G. Mu *et al.*, npj Quantum Materials **6**, 55 (2021)].

Based on these results from newly ***refined multi-peak fit method***, we plot the temperature-dependent energy shifts for three distinct peaks. Notably, we clearly observe the consistent energy shifts towards lower binding energies across the all the peaks. ***These findings align with our previous results, reinforcing the robustness of our initial statement regarding the hole doping nature in NblrTe₄.***

In response to the reviewer’s question about the resolution, we would like to clarify that the instrument resolution, indicated as “20 meV”, refers to the convolution of the original spectrum with a Lorentzian function featuring a full width half maximum (FWHM) of 20 meV. It is crucial to emphasize that while this value can cause broadening, but it does not represent the smallest energy scale that can be resolved [A. Damascelli, Phys. Scr. **T109**, 61-74 (2004)]. In our measurements, we have conducted a scan of the E - k dispersion with an energy step of 6-8 meV, a significantly finer value than the resolution

of 20 meV. This finer step ensures the validation of our data analysis. Indeed, the peaks observed in our experiment exhibit a FWHM of around 100 meV, separated by 20 meV. This condition has proven to be sufficient for accurately capturing the peak positions, as demonstrated in Figure R3 (a). Therefore, our methodology enables the effective capture and analysis of energy shifts at a scale as small as 20 meV.

Furthermore, our argument is supported by a reference, as illustrated in Figure R3 (b) [S. -D. et al., *Nature* **601**, 562-567 (2022)]. This figure shows the temperature-dependent superconductor gap obtained from k_F EDCs, as detailed in the schematic on the right side. The gap sizes were determined from k_F EDC at various momentum points, marked by distinct Fermi surface angles in the top-right inset, accompanied by corresponding resolution details in the bottom-right corner. The difference in gap size between low and high temperatures indicates the energy shifts (ΔE) of quasiparticle peaks. Importantly, this graph clearly shows that comparable or small energy shifts with respect to resolution can be adequately captured, especially at $T < T_c$. Therefore, we find that our energy shifts closely align with those presented in the reference.

We appreciate the reviewer for once again suggesting more accurate and definitive fitting method. Thanks to the comment, we have been able to enhance the precision of our analysis, confidently supporting our conclusion. In response to the reviewer's comment, we have revised Figs. 3 (d) and (e) and the main text accordingly in the latest version of our manuscript (page 8-9 lines 151-154, page 9 lines 173, page 9 lines 176-178). In addition, Figure R1 and R2 were included in Figs. S9 of SI.

Figure R3. Resolution in ARPES experiment. (a) Illustration of two Lorentzian functions with a full width half maximum (FWHM) of 100 meV, depicted in red and blue, and separated by a 20 meV energy shift. (b) Relevant reference demonstrating energy shifts comparable to the corresponding resolutions, extracted from [S.-D. et al., *Nature* **601**, 562-567 (2022)]. It illustrates the gap size of a high T_C superconductor as a function of temperature, with the resolution denoted by vertical bars in the bottom-right corner. Gap sizes were determined from k_F EDCs at various momentum points, marked by distinct Fermi surface angles in the top-right inset. In the schematic on the right side, the superconductor gaps are defined by the position of the quasiparticle peaks from the E_F , also highlighting the energy shift (ΔE) between low and high temperatures.

2) To investigate surface dependence, the authors provided three different Fermi surfaces together with theoretical calculations. They claimed that a small hole pocket at the Y point only exists on the S1 surface, therefore, all the displayed Fermi surfaces were expected to be from the same surface. However, to me, it is unclear whether they observed the same surface. I do not see clear periodicity along the k_y axis as marked in the figure, and I am even unconvinced if the “pockets” are really at the Y point. Additionally, all three plots seem very different from each other, and even the second one more closely resembles Fermi surface from S2 (see Fig. 3(c) in Ref. 29 of the main text). As non-linear Hall effect is from surface band, it is very crucial to unambiguously determine surface termination. For the theoretical parts, I agree that -0.3 eV feature similarly shifts as a function of carrier concentrations. However, it is from bulk which is not expected to contribute Berry curvature dipole. Can they apply the argument to surface states (i.e. states lack of glide mirror symmetry) near the Fermi level?

Our reply: We agree with the reviewer's observation that our previous reply may not have sufficiently demonstrated the determination of Fermi surfaces originating from specific terminated surfaces. To facilitate a more lucid examination of the surface termination, we have overlaid illustrations of the main features of S1 and S2 terminations (Figure R4). The overlaid results suggest that the Fermi surface's main features align more closely with those of the S1 termination rather than the S2 termination.

The double-elliptical hole pockets near the Y point in Figure R4 (a) and the single-elliptical hole pocket near the Γ point in Figure R4 (b) are the signature features of S1 and S2 termination, respectively. A crucial and consistent observation is that all three measured Fermi surfaces have a double-elliptical hole pockets near the Y point. Although the second Brillouin zone, marked in the green box of Fermi surface (ii), exhibits lower intensity, the remaining sections demonstrate good agreement with the main features of the S1 termination. Therefore, we suggest that the unbalanced intensity of second Fermi surface (ii) may be attributed to the matrix element effect.

Figure R4. Fermi surfaces of NbIrTe₄. (a) The Fermi surfaces overlaid on illustration of the double-elliptical hole pockets near Y point, existing only in S₁ terminated surface. (b) The Fermi surfaces overlaid on the illustration of the single-elliptical hole pocket near the $\bar{\Gamma}$ point, existing only in S₂ terminated surface. Experiments (i), (ii), and (iii) represent three independent samples measured at temperature of 20 K, 20 K, and 10 K, respectively. (c) Two different terminated surfaces (S₁ and S₂) in NbIrTe₄ reported from ref.29 [S. A. Ekahana et al., *Phys. Rev. B* **102**, 085126 (2020)]. Black dashed boxes highlight the characteristic features of each terminated surface.

Despite the presence of intensity at the $\bar{\Gamma}$ point, particularly in the second Fermi

surface (ii), it remains challenging to definitively conclude whether this intensity originates from the single-elliptical hole pocket near the Γ point in S2 termination or the intensity near the Γ point in S1 termination. Therefore, to the best of our examination, we propose that the presence of an elliptical hole pocket near the Y point can be evidence for the manifestation of the S1 terminated surface feature.

We also agree with the reviewer that the Berry curvature dipole is mainly contributed from both the top and bottom surfaces in which the glide mirror symmetry of the bulk $Pmn2_1$ space group is not present. Therefore, regardless of the specific termination, both the topmost and bottommost layers are anticipated to exhibit surface nonlinear Hall effect, as explained in Figure R5. This statement aligns with this detailed in ref. 6 [D. Kumar et al., Nat. Nanotech. **16**, 421-425 (2021)].

Figure R5. Schematic illustration representing the contribution of Berry curvature dipole (D_a) of $NblrTe_4$ based on surface or bulk system, extracted from ref. 6 [D. Kumar et al., Nat. Nanotech. **16, 421-425 (2021)].** Both topmost and bottommost layers have a single mirror plane M_a , expecting to exhibit nonlinear Hall effect from surface while bulk system has the mirror ($\{M_a|(0,0,0)\}$) and glide mirror ($\{M_b|(1/2,0,1/2)\}$) resulting in zero D_a .

Figure R6. Electronic structures of six-layer slab geometry for undoped and 0.025 h/f.u. doped NbIrTe₄. Slab band structures of (a) undoped and (b) doped NbIrTe₄. (c) Comparison between the undoped (black lines) and doped (magenta lines) band dispersions around the Fermi energy. (d) Atomic configuration of the six-layer slab. The blue, green, and red boxed regions represent the top surface layer, middle layers, and bottom surface layer, respectively. (e-g). Band structures of the undoped slab projected on the atoms in the top surface layer (panel e), middle layers (panel f), and bottom surface layer (panel g). (h-j). Band structures of the doped slab projected on the atoms in the top surface layer (panel h), middle layers (panel i), and bottom surface layer (panel j).

Lastly, we agree with the reviewer that the chemical potential shift might be different in these regions. A way to check this possibility is to investigate the band structures of the slab geometry, which are projected onto the states near the surface regions, and check whether there are notable changes in the projected band structures with respect to the chemical potential shift.

We calculate the electronic structures of the six-layer slab with undoped and doped (0.025 h/f.u.) cases. Figure. R6 (a) and (b) present the band structures of the undoped and doped six-layer slab geometry, respectively. We find that the effect of doping is mainly the rigid band shift of about 20 meV as shown in the comparison around the Fermi energy (panel (c)), which is consistent with our bulk calculations.

In addition, we evaluate the projected bands on the atoms in the top and bottom layers and their doping dependence Figure R6 (d) shows the atomic configuration of the six-

layer slab, where we divide the atomic structure into the top surface layer, middle bulk-like layers, and bottom surface layer, shown in blue, green, and red boxes, respectively. Figure R6 (e), (f), and (g) are bands projected on the atoms belonging to the top, middle, and bottom layers, respectively, and similarly, Figure R6 (h-j) are corresponding atom-projected bands for the doped case. We find no significant differences between the projected bands among the three regions, consistent with the relatively weak inter-layer van der Waals coupling. More importantly, we find no noticeable changes in the projected bands with respect to doping, suggesting that the main effect of doping is a rigid shift in the band structures. Thus, we expect that the states around the top and bottom surfaces mainly contributing to the BCD are shifted in energy as carrier concentration changes.

Following the reviewer's comment, we have added a sentence, "Reducing the number of electrons corresponding to a 0.025 h/f.u. doping induces the chemical potential shifts of about -20 meV both for bulk and slab geometry (see Fig. S13 and S14 in SI).", to the page 12-13 lines 251-253 and added the Fig. R6 (Fig. S16) to SI.

3) Fittings for EDC at the C point (Fig. R7) look ambiguous as they obviously deviate near the Fermi level. This should be improved by convoluting FD and experimental resolution with Lorentzian curves. At this moment, both the short-range and long-range fittings provide no information. For instance, I can randomly set a "cut-off" energy where fitting results (black curve) start departing from experimental observation. This ultimately alters peak positions, especially peaks near the Fermi level and -0.3 eV, found from the fittings. Moreover, the C point is where a band dispersion crosses the Fermi level (Fermi momentum), but in the fittings I do not see any quasiparticle peak at the Fermi level even at 10 K.

Our reply: We fully agree with the reviewer's suggestion that our analysis can be further improved by multiplying the FD function and convoluting experimental resolution with Lorentzian curves, ranging from the Fermi energy (E_F) to higher binding energies. As highlighted in our response to the #1 question, **the fitting with convoluted Lorentzian curves (multi-peak fit approach)** is addressed at both the Γ and C points. This method significantly improves the fitting quality of the peaks near the Fermi level and -0.3 eV,

eliminating deviations between the data and fit by convoluting FD function and experimental resolution with Lorentzian curves.

This improvement significantly enhances the quality of our manuscript, reinforcing our conclusions on chemical potential shifts towards higher binding energies. It confidently supports the inherent tendency of hole doping nature in NbIrTe_4 , aligned with our initial statement.

We also agree with the reviewer's observation that the EDCs at the C point lack a distinct quasiparticle peak compared to the Fermi surface presented in the main manuscript (Fig. 3b). The sample that we performed temperature dependent measurement at the C point is not identical to the one used for electronic structure analysis in Fig. 3a-c of the manuscript. The presence of the quasiparticle peak at the C point could be influenced by factors such as cleaving conditions or matrix element effects. This observation agrees with the result reported in ref. 29 [S. A. Ekahana et al., Phys. Rev. B **102**, 085126 (2020)], where a similar absence of a distinct quasiparticle peak. Additionally, our intention was to select the most well-defined peak for the multi-peak fitting analysis, rather than choosing the k_F EDCs from the C point.

We recognize that the readers may have similar concern regarding the electronic structure of C point. In response of this, we have deleted “from panel b” in caption of Fig. 3 (d) as detailed on page 9 line 157.

Figure R7. Two independent ARPES experiments of NbIrTe₄. (a-b) Fermi surface and E - k dispersion along the Y-S direction. Experiment (a) involved temperature-dependent measurements, specifically to investigate the C point, while experiment (b) focused on the basic electronic band structure as presented in Fig. 3a-c of the main manuscript. The red dashed lines and arrows indicate the position of $k_x = 0.13 \text{ \AA}^{-1}$ and C point.

Reply to Reviewer #3

Comments:

The authors have provided more benchmark calculations and addressed all my comments. I recommend its publication, while there are some loose ends to fix in the manuscript. For example,

the related reference, Nano letters 17, 467-475 (2017) that predicted the existence of Weyl points the momentum-dependent Berry curvatures in these materials, is not added in the manuscript as said in the rebuttal letter.

Our reply: We sincerely express the gratitude to the reviewer for recognizing “*more benchmark calculations*” and recommending the publication of our work in *Nature communications*. Additionally, we acknowledge that we missed the reference of *Nano Lett. 17, 467-475 (2017)* in main manuscript. Thank you for bringing this to our attention. Following the reviewer’s suggestion, we have added this reference, as detailed on page 9 line 169.

List of changes

Changes in the Manuscript

1. In the Manuscript, new figure **3 d** and **e** has been updated along with corresponding its caption.
2. In the Manuscript pages 8-9 lines 151-154, the sentence “along with black fitted...respectively” has been modified.
3. In the Manuscript page 9 line 169, the reference number 31 “*Nano Lett. 17, 467-475 (2017)*” has been added.
4. In the Manuscript page 9 line 173, the sentence “the two peaks...-1.0 eV” has been modified.
5. In the Manuscript page 9 lines 176-178, the sentence “obtained from...as detailed in Fig. S9.” has been modified.
6. In the Manuscript page 12 lines 251-253, the sentence “Reducing the number (see Fig. S14 in SI)” has been modified.
7. In the Manuscript page 16 lines 346-347, the phrase “RS-2023-00284081” has been added.
8. In the Manuscript page 20 lines 441-442, the reference “31. Liu, J. et al..... (2017).” has been added.

Changes in the Supplementary Information

1. In the Supplementary Information, figure **S6** have been revised along with corresponding its caption.
2. In the Supplementary Information, a whole new section titled “**Multi-peak fit analysis of temperature-dependent EDCs**” along with a new figure **S9** have been added.
3. In the Supplementary Information, a whole new section titled “**Doping-dependent electronic structure for the slab geometry**” along with a new figure **S14** have been added.

REVIEWER COMMENTS

Reviewer #2 (Remarks to the Author):

The authors now provided better visualization as well as more comprehensive support for their argument. I had particularly recommended improving discussion regarding ARPES, and most of the revised parts work fine in that sense. Therefore, the manuscript now has my endorsement for publication if the last concern described below is resolved.

I acknowledge that the authors performed refined peak fittings for energy distribution curves, while the fitting results for the EDCs at C are not very promising. For Gamma the results look OK. I would still like to recommend removing fitting results for EDCs at C from the paper (i.e. Fig. 3e) as they are unreliable (the authors also mentioned this in the last rebuttal letter). The absence of quasiparticle peak at C in Fig. R7a even makes the measurement at C more concerning.

Manuscript ID: NCOMMS-23-08582C

Title: Spin-orbit-splitting-driven nonlinear Hall effect in NbIrTe₄

Authors: Ji-Eun Lee; Aifeng Wang; Shuzhang Chen; Minseong Kwon; Jinwoong Hwang; Minhyun Cho; Ki-Hoon Son; Dong-Soo Han; Jun Woo Choi; Young Duck Kim; Sung-Kwan Mo; Cedomir Petrovic; Choongyu Hwang; Se Young Park; Chaun Jang; Hyejin Ryu

We sincerely thank the reviewer for his/her comments and suggestion, which have significantly improved the quality of our manuscript. Below is our point-by-point response to the reviewer's comment. The reviewer's comment is shown in italics in the order that appeared on the report, followed by our response in a sans-serif font. The manuscript has been revised according to the reviewer's advice, summarized in the list of changes.

Reply to Reviewer #2

Comments:

The authors now provided better visualization as well as more comprehensive support for their argument. I had particularly recommended improving discussion regarding ARPES, and most of the revised parts work fine in that sense. Therefore, the manuscript now has my endorsement for publication if the last concern described below is resolved.

Our reply: We sincerely express our gratitude to the reviewer for recognizing “**the better visualization**” and “**more comprehensive support**” for our revised manuscript. Furthermore, we extend our sincere thanks for the **positive decision**. Following the reviewer's comment, we have revised the manuscript as detailed below.

I acknowledge that the authors performed refined peak fittings for energy distribution curves, while the fitting results for the EDCs at C are not very promising. For Gamma the results look OK. I would still like to recommend removing fitting results for EDCs at C from the paper (i.e. Fig. 3e)

as they are unreliable (the authors also mentioned this in the last rebuttal letter). The absence of quasiparticle peak at C in Fig. R7a even makes the measurement at C more concerning.

Our reply: We agree with the reviewer's comment regarding the EDCs at the C point. Following the reviewer's suggestion, we have removed the fitting result for EDCs at C from Figure 3(e) (see Figure R1) and updated Figure 3(e) to Figure R1 with corresponding changes in the figure caption and the main text. We have also removed Fig. S7 of the previous version of Supplementary Information, which shows the multi-peak fitting results at the C point.

Figure R1. Temperature dependence of the energy shifts ΔE (left axis) are taken from the peak position shift of EDCs at Γ ($\Delta E = E - E_{T=20\text{K}}$) point obtained by multi-peak fit. Hole carrier density (n , right axis) is obtained from the reported Hall measurement²⁸.

List of changes

1. Figure 3e has been modified by removing the data from the energy distribution curves at the C point, following Reviewer #2's suggestion, with the corresponding changes to the caption and main text (page 9 line 156 and 175).

2. Figures **S8** and **S9b** on the previous submitted SI, which discuss the fitting at the C-point, have been removed following Reviewer #2's suggestion, with the corresponding changes to the caption.
3. We have found a minor error in the figure numbering in the previously submitted SI. The figure that comes after Figure **S7** was numbered as Figure **S9**, which should have been Figure **S8**. This error has been corrected.
4. Following changes #2 and #3, all the references to the supplementary figures have been double-checked and corrected.